# Unveiling multifunctional synthetic boundaries for enhanced mechanical and electrochemical performance in densified thick composite electrodes

Bo Nie[1,6], Seok Woo Lee [1,6], Ta-Wei Wang[1], Tengxiao Liu[2], Ju Li [3,4] ✉ & Hongtao Sun [1,2,5] ✉

High energy density lithium-ion batteries are essential for sustainable energy solutions, as they reduce reliance on fossil fuels and lower greenhouse gas emissions. Increasing electrode thickness is an effective strategy to raise energy density at the device level, but it poses inherent scientific challenges. Thick electrodes typically require a highly porous structure (over 40% porosity) to maintain sufficient charge transport. Such porosity sharply lowers volumetric energy density, limiting use in space-constrained applications. Conversely, direct densification of thick electrodes intensifies charge diffusion limitations and exacerbates mechanochemical degradation. To overcome these trade-offs, we explore a geology-inspired, transient liquid-assisted densification process that produces dense, thick electrodes with multifunctional synthetic secondary boundaries. These boundaries provide three key benefits: (1) strain resistance that mitigates mechanochemical degradation, as demonstrated by operando full-field strain mapping; (2) enhanced charge transport across boundary phases in thick and dense electrodes (thickness > 200 μm, relative density > 85 %), leading to improved comprehensive electrochemical performance with a volumetric capacity of 420 mAh cm$^{-3}$, an areal capacity of 23 mAh cm$^{-2}$, and a specific (gravimetric) capacity of 195 mAh g$^{-1}$ at a current density of 1 mA cm$^{-2}$; and (3) tailored conducting phases that increase active material content to 92.7 % by weight, further elevating volumetric capacity to 497 mAh cm$^{-3}$.

High-energy-density secondary batteries are crucial for meeting the growing demands of portable electronics, electric vehicles, and the integration of renewable energies such as solar and wind into the electricity grid. These rechargeable batteries play a critical role in promoting sustainable energy solutions by replacing fossil fuels and reducing greenhouse gas emissions, therefore advancing the transition toward a decarbonized economy. Increasing electrode thickness emerges as a viable strategy for boosting energy density at the device

[1]The Harold & Inge Marcus Department of Industrial & Manufacturing Engineering, The Pennsylvania State University, University Park, PA, USA. [2]Department of Biomedical Engineering, The Pennsylvania State University, University Park, PA, USA. [3]Department of Nuclear Science and Engineering, Massachusetts Institute of Technology, Cambridge, MA, USA. [4]Department of Materials Science and Engineering, Massachusetts Institute of Technology, Cambridge, MA, USA. [5]Materials Research Institute (MRI), The Pennsylvania State University, University Park, PA, USA. [6]These authors contributed equally: Bo Nie, Seok Woo Lee. ✉e-mail: liju@mit.edu; hongtao.sun@psu.edu

level (Fig. 1a). For instance, raising the electrode thickness from 20 to 200 micrometers can increase the active material loading by more than 30% across the entire device, improving the overall specific energy (gravimetric energy density)[1]. Nevertheless, this thick electrode strategy poses critical challenges due to sluggish charge transport kinetics, which lead to substantially reduced battery performance.

Recent advancements in developing thick electrodes focus on enhancing charge transport by strategically arranging internal pores to reduce their tortuosity[2–14]. Although these architected thick electrodes achieve high specific energy, their high porosity (over 40%) drastically reduces volumetric energy density. Practical electrodes need to maintain high energy density both gravimetrically and volumetrically, making them suitable for mass- and space-constrained applications, such as mobile transportation. However, densifying thick electrodes results in dramatically increased charge transport resistance. Moreover, anisotropic straining of active material building blocks, such as secondary polycrystalline particles, amplifies mechanochemical degradations in overall thick electrodes, further exacerbating battery performance decay[15]. Therefore, achieving both high gravimetric and volumetric battery performance in densified thick electrodes remains a substantial scientific challenge.

Furthermore, densifying thick composite positive electrodes containing large amounts of ceramic active material (e.g., $LiNi_{0.8}Mn_{0.1}Co_{0.1}O_2$) as a matrix typically requires energy-intensive sintering processes. Current state-of-the-art sintering technologies, such as liquid phase sintering, field-assisted sintering technology, fast-firing sintering, and laser sintering[16], can achieve high densities but require high sintering temperatures (800–2000 °C), which are not feasible for incorporating materials like polymer binders and carbon additives that degrade at much lower temperatures (<400 °C). Thus, the substantial discrepancy in processing temperatures between ceramic active materials and inactive additives poses a considerable processing challenge for making highly densified composite electrodes.

To overcome both scientific and processing challenges, we introduce a geology-inspired densification process via pressure solution creep. This method creates localized solvothermal micro-environments between ceramic particles by applying uniaxial pressure, moderate heating (120 °C), and a small amount of transient liquids (Fig. 1b and Supplementary Fig. 1)[17]. Within these confined environments, additive materials (e.g., polymers, lithium salts) partially dissolve at compressed surfaces and subsequently precipitate onto pore surfaces, driven by transient liquid-assisted mass transfer (Fig. 1b)[18–21]. This approach dramatically lowers the activation energy and processing temperature compared to traditional solid-state densification processes along dry surfaces/interfaces[22–24]. Crucially, the formation of a secondary boundary phase within the densified thick composite electrodes offers several advantages. It enhances damage tolerance of composite electrodes by mitigating mechanochemical degradations during electrochemical cycling, and it facilitates efficient charge transport, enabling improved trade-off battery performance in thick, densely packed composite electrodes (Fig. 1b).

## Results

### Pressure solution creep-induced densification of composites

In this work, we integrate $LiNi_{0.8}Mn_{0.1}Co_{0.1}O_2$ (NMC811) secondary particles with polymer, ionic liquid (IL), and carbon additives into a densified composite electrode (Supplementary Fig. 2). Specifically, a lithium bis(trifuloromethylsulfonyl)imide (LiTFSI) as an additional Li salt and poly(vinylidene fluoride-co-hexafluoropropylene) (PVDF-HFP) polymer are dissolved in a miscible solution of 1-Ethyl-3-methylimidazolium bis(trifluoromethylsulfonyl)imide (EMIMTFSI) ionic liquid (IL), acetone, and dimethylformamide (DMF), creating a poly(ionic liquid) mixture. During the densification process, this solution mixture, which includes DMF-acetone dual transient liquids, transports soluble species (e.g., LiTFSI and PVDF-HFP) along with insoluble carbon additives (e.g., graphene and carbon nanofiber (CNF)) from the compressed surfaces of NMC811 secondary particles (dissolution zones) to the non-contacting surfaces via stress-driven

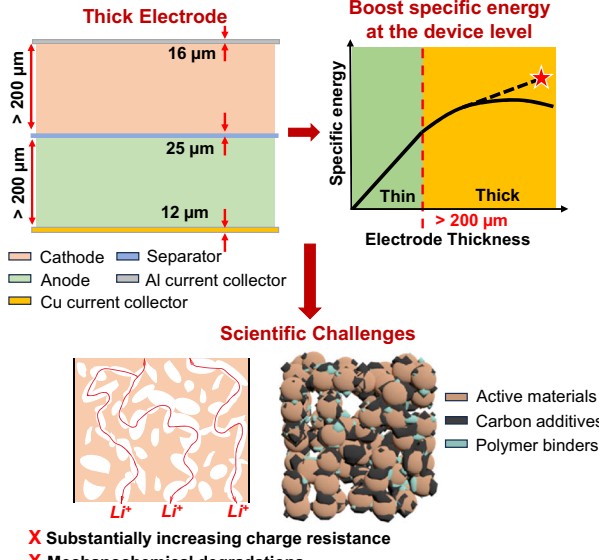

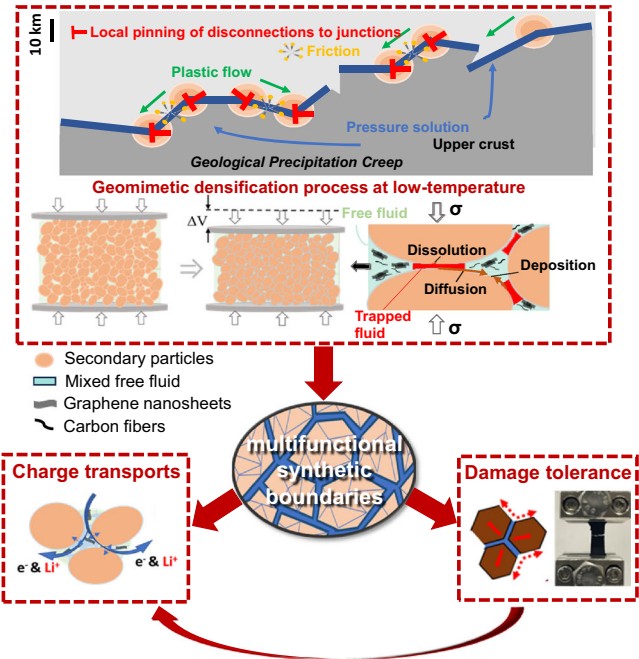

**Fig. 1 | Schematic depicting opportunities, challenges, and solutions for developing thick composite electrodes. a** Opportunities and challenges for thick electrode design. **b** Engineering multifunctional synthetic boundaries in densified thick composite electrodes for advancing battery technologies. The labels 12 µm, 16 µm, and 25 µm indicate the typical thicknesses of the Cu current collector, Al current collector, and separator, respectively.

mass transfer (Fig. 1b). As the processing temperature gradually increases to 120 °C, the DMF (flash point: 58 °C) and acetone (boiling point: 56 °C) transient liquids evaporate, leading to the concentration and precipitation of a supersaturated poly(ionic liquid) gel (PILG) phase on pore surfaces (deposition zones). Consequently, the locally Li$^+$-enriched PILG, along with graphene and CNF additives, forms a secondary boundary phase that integrates the NMC811 secondary particles into a densified composite.

The structures and phases of densified composites involving various liquids (e.g., acetone, DMF, and IL) during the densification process were characterized by X-ray diffraction (XRD) and Fourier transform infrared spectroscopy (FT-IR). The crystal structure of NMC811 remained unchanged in the composites after densification (Supplementary Fig. 3), highlighting the advantage of this low-temperature processing method, which prevents side reactions. FT-IR spectra confirmed the presence of the PILG phase, with peaks at 1339 cm$^{-1}$ and 1198 cm$^{-1}$ corresponding to the characteristic TFSI$^-$ for EMIMTFSI IL (Supplementary Fig. 4)[25]. The peak at 867 cm$^{-1}$ and the bands between 794 and 840 cm$^{-1}$ correspond to the polar β phase of PVDF-HFP, which offers improved ionic conductivity compared to the non-polar α phase[26–28] Thermogravimetric analysis (TGA) confirmed that the NMC811 content is 73.9 wt% in the densified composites with IL and 81.0 wt% in those without IL (Supplementary Fig. 5).

## Transient liquid-regulated interface engineering for enhanced damage tolerance

In our composite system, transient liquids play a crucial role in the densification process. To investigate their behaviors, we measured the viscosity of PVDF-HFP polymer dissolved in various liquids as the temperature increased from 25 °C to above 110 °C (Supplementary Fig. 6), simulating the heating conditions during densification. The results revealed an increase in shear viscosity at 60 °C for PVDF-HFP dissolved in DMF. When the polymer was dissolved in a mixture of DMF and acetone, the viscosity began to rise at a lower temperature because of acetone's low boiling point of 56 °C, and it increased more sharply above 60 °C, promoting more efficient evaporation. The addition of IL to the mixture resulted in higher initial viscosity at lower temperatures, which gradually decreased as the temperature rose. This improved fluidity with increasing temperature likely facilitates mass transfer, helping to fill unoccupied voids during the densification process. As a result, the composite densified using dual transient liquids and IL achieved the highest relative density of 85.5%, compared to other counterparts (Supplementary Fig. 7, and associated calculations for relative density and porosity in Supplementary Text). Conversely, the densification process conducted without any transient liquids and IL, referred to as hot pressing, resulted in the lowest relative density of 70.0%. Additionally, real-time monitoring of processing conditions such as temperature, pressure, and linear shrinkage revealed the densification kinetics (Supplementary Fig. 8).

To further evaluate the effectiveness of the densification process assisted by different liquid aids, we compared the mechanical properties of these densified composites. The hot-pressed pellet, made without using any liquids, barely withstood mechanical force during the tensile test. However, adding transient liquids such as DMF, acetone, and their mixture resulted in simultaneous improvements in ultimate tensile strength (UTS), elastic modulus, and material toughness (defined as energy absorption before rupture, represented by the area under the stress-strain curve) of the densified composites (Fig. 2a–c). Optimizing the transient liquids from DMF to a DMF-acetone mixture led to a more than sevenfold increase in material toughness (from 1770 to 14060 J m$^{-3}$) and a 300% increase in UTS (from 1.26 to 5.15 MPa). These enhancements indicate a significantly improved synergy between UTS and material toughness (Fig. 2c). Incorporating IL into the composite system formed a more ductile PILG secondary boundary phase, further enhancing material

toughness (e.g., 22850 J m$^{-3}$) while causing a slight reduction in UTS (e.g., 4.49 MPa) due to the plasticizing effect of the IL[29,30]. The integration of brittle ceramic secondary particles with a ductile boundary phase mimics the brick-and-mortar structure in nacre, which is toughened through cooperative plastic deformation[31]. Additionally, after soaking the six different composites in organic electrolyte for 48 h, the two densified composites treated with both acetone and DMF as transient liquids—namely, NMC811-PVDF-HFP (without IL) and NMC811-PILG (with IL)—exhibited more robust structural integrity compared to the other samples, which showed noticeable swelling in the electrolyte (Supplementary Fig. 9).

To understand the improved damage tolerance, we conducted real-time full-field strain mapping using digital image correlation (DIC) during tensile testing, examining the NMC811-PILG composite (Fig. 2d, e) and the NMC811-PVDF-HFP composite (Supplementary Fig. 10). Localized strain concentrations, indicated by red-colored regions in the Y-direction strain mapping (Fig. 2d), revealed the deformation of soft domains, such as the PILG phase, within the NMC811-PILG composite. In the X-direction strain mapping, more soft domains exhibited positive strains (red-colored regions) rather than negative strains (blue-colored regions) (Fig. 2e). These site-specific strain responses were further confirmed by X- and Y-strain profiles along a selective pathway (Fig. 2f), indicating abundant local sites with negative Poisson's ratios $\left(\nu_y = -\dfrac{\varepsilon_x}{\varepsilon_y}\right)$. This negative Poisson's ratio effect in soft domains is likely influenced by the densification process, which tailors the interaction forces (such as friction and adhesion) between local stiff (integrated secondary particles) and soft domains, as well as the cohesion strength within the boundary phase[32,33]. The failure of composite materials preferentially occurs at local regions where strain concentration cannot be fully accommodated. Therefore, the soft domains with negative Poisson's ratios may mitigate the complete exhaustion of local strains, resulting in enhanced material toughness. Additionally, the localized stress-strain correlations of two representative regions were extracted from a series of time-resolved strain mappings, exhibiting brittle (region 1) and ductile (region 2) characteristics, respectively (Fig. 2g). Consequently, transient liquid-regulated interface engineering via a low-temperature densification process provides a viable solution for integrating brittle ceramics and ductile boundary phases into a densified composite, significantly improving the synergy between strength and material toughness.

## Morphological and compositional characterizations of structured composites with tailored secondary particle-to-boundary phase ratios

A key advantage of our transient liquid-assisted densification approach is its ability to seamlessly integrate dissimilar components, such as inorganic secondary particles and the organic PILG phase, into highly dense composite architectures at low processing temperatures. Unlike conventional fabrication methods for organic–inorganic composites, which often result in low inorganic content, our strategy enables the incorporation of a high weight percentage of NMC811 secondary particles. To explore this capability, we investigated a series of NMC811-PILG composites with increasing NMC811 content: 73.9 wt%, 86.7 wt%, and 92.7 wt% as evidenced by TGA (Supplementary Figs. 11, 12, and Tables 1–4).

Morphological and compositional characterizations were performed using scanning electron microscopy (SEM) and energy-dispersive X-ray spectroscopy (EDS) from both top and cross-sectional views (Fig. 3a–j). The SEM images reveal progressively more compact integration of secondary particles with increasing NMC811 content. EDS mapping provides spatial distribution of characteristic elements from NMC811 (e.g., Ni, Co, O) and the PILG boundary phase (e.g., C and F, attributed to carbon additives, polymers, and IL components) in the top-view images (Fig. 3a, b, d, e, g, h,

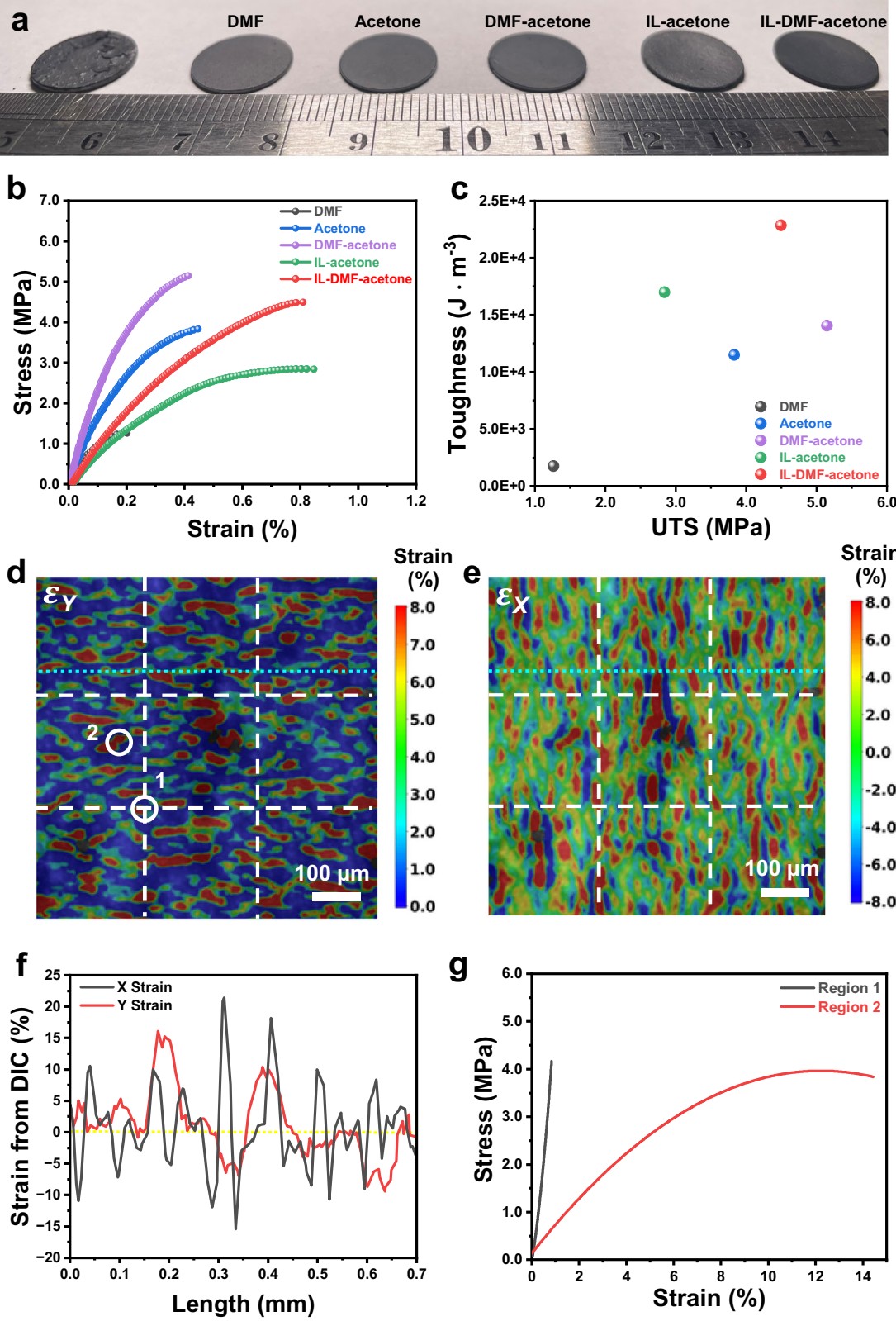

**Fig. 2 | Mechanical characterizations of densified composites. a** Digital images of pellets densified using various liquids. **b** Stress-strain curves of the composites prepared using various liquids. **c** Correlations between material toughness and ultimate tensile strength exhibiting enhanced damage tolerance. Real-time full-field strain mapping of the NMC811-PILG composite in the Y-direction (**d**) and X-direction (**e**) obtained via DIC analysis at a global strain of 0.80% during uniaxial tensile loading along the Y-direction. **f** Strain profiles along the selective linear pathway (aqua dotted line) in (**d**) and (**e**). **g** Localized stress-strain curves of two representative regions derived from a series of time-resolved strain mappings. Source data are provided as a Source Data file.

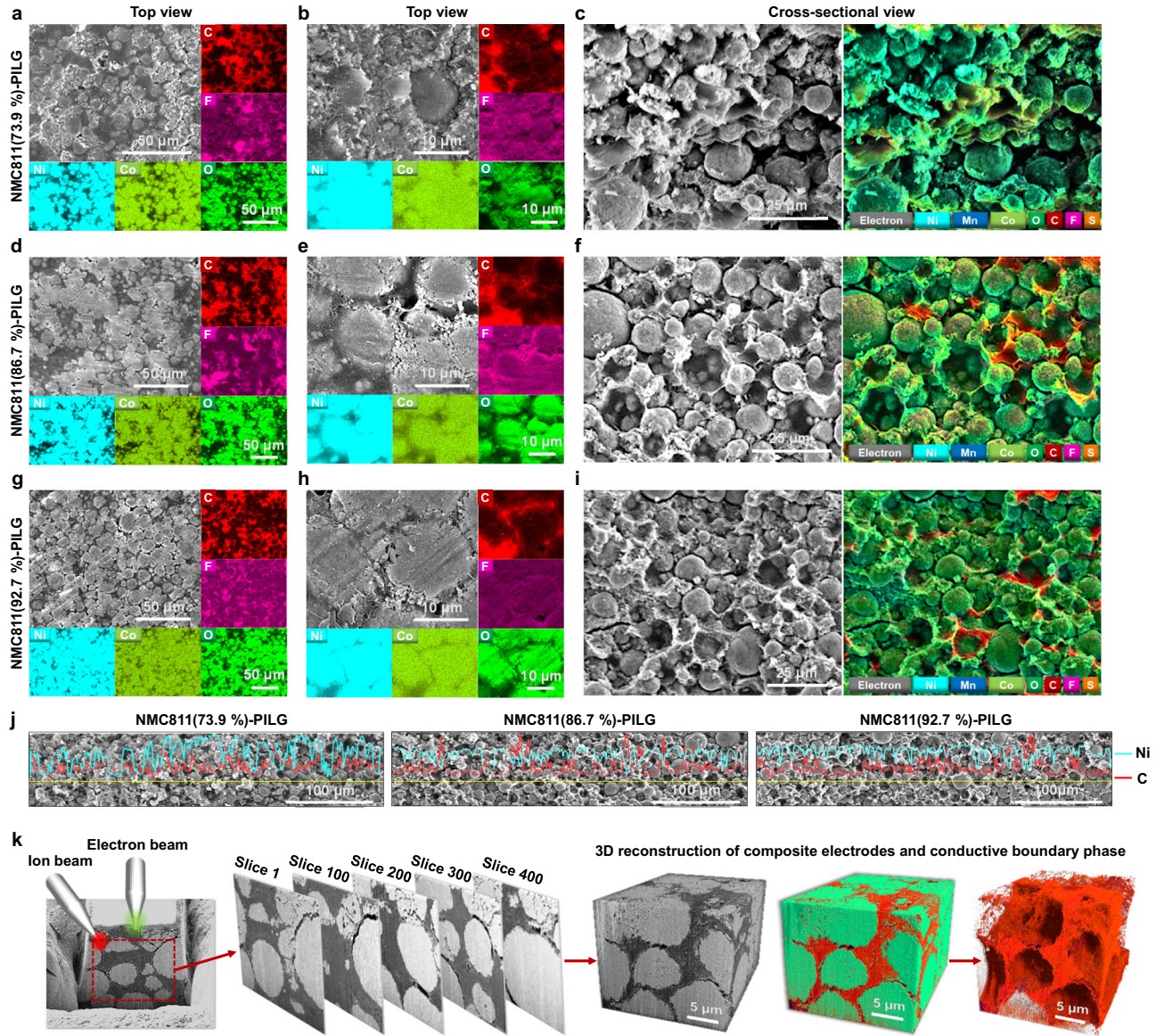

**Fig. 3 | Morphological and compositional characterizations of composites densified via transient liquid-assisted processing.** Top-view and cross-sectional view of densified composites, along with corresponding elemental mapping of C, F, Ni, Co, and O obtained by EDS for NMC811(73.9%)-PILG (**a**–**c**), NMC811(86.7%)-PILG (**d**–**f**), and NMC811(92.7%)-PILG (**g**–**i**). **j** EDS line scan profiles acquired from fractured cross-sections, showing the spatial distribution of Ni and C elements across

secondary particles and synthetic boundary phases. **k** Demonstration of FIB-sliced SEM images and their corresponding 3D reconstructions, illustrating the integrated multi-phase composites and segmented conductive boundary phase (green-colored regions: NMC811 secondary particles; red-colored regions: PILG conducting phase). Source data are provided as a Source Data file.

and Supplementary Fig. 13). Furthermore, elemental overlay maps from the cross-sectional views were used to correlate structural features with localized chemical composition, clearly highlighting the uniform distribution of elements within the NMC811 particle regions and the surrounding synthetic boundary phases (Fig. 3c, f, i, and Supplementary Fig. 14).

To further examine the microstructural evolution induced by the transient liquid-assisted densification process, EDS line scans were conducted across a 400 μm region in the cross-sections of NMC811-PILG composites with varying active material contents (Fig. 3j). In the composite with 73.9 wt% NMC811, the Ni signal exhibits relatively wide interparticle spacing, while a pronounced C signal is detected within the gaps, suggesting the presence of well-distributed conducting boundary phases in these interstitial regions. As the NMC811 content increases to 86.7 wt% and 92.7 wt%, the Ni signal becomes increasingly continuous, reflecting a more densely packed particle network.

Concurrently, the C signal remains evident but appears more confined to narrower regions, indicating the formation of a thinner, yet well-integrated conducting boundary phase.

To quantitatively analyze the three-dimensional (3D) architecture of our densified composites, focused ion beam–scanning electron microscopy (FIB-SEM) was performed on a representative NMC811-PILG composite containing 86.7 wt% active material content and exhibiting a relative density of 87.6% (as estimated by rough calculation, see Supplementary Text). A total of 400 serial cross-sectional sliced images were acquired and reconstructed into a 3D volume (28 μm × 18 μm × 20 μm) (Fig. 3k). Owing to the distinct grayscale contrast between different phases, voxel-level segmentation enabled identification of secondary NMC811 particles, the conducting PILG phase, and residual pores[34].

The reconstructed 3D morphology reveals a highly interconnected network of the conductive boundary phase percolating

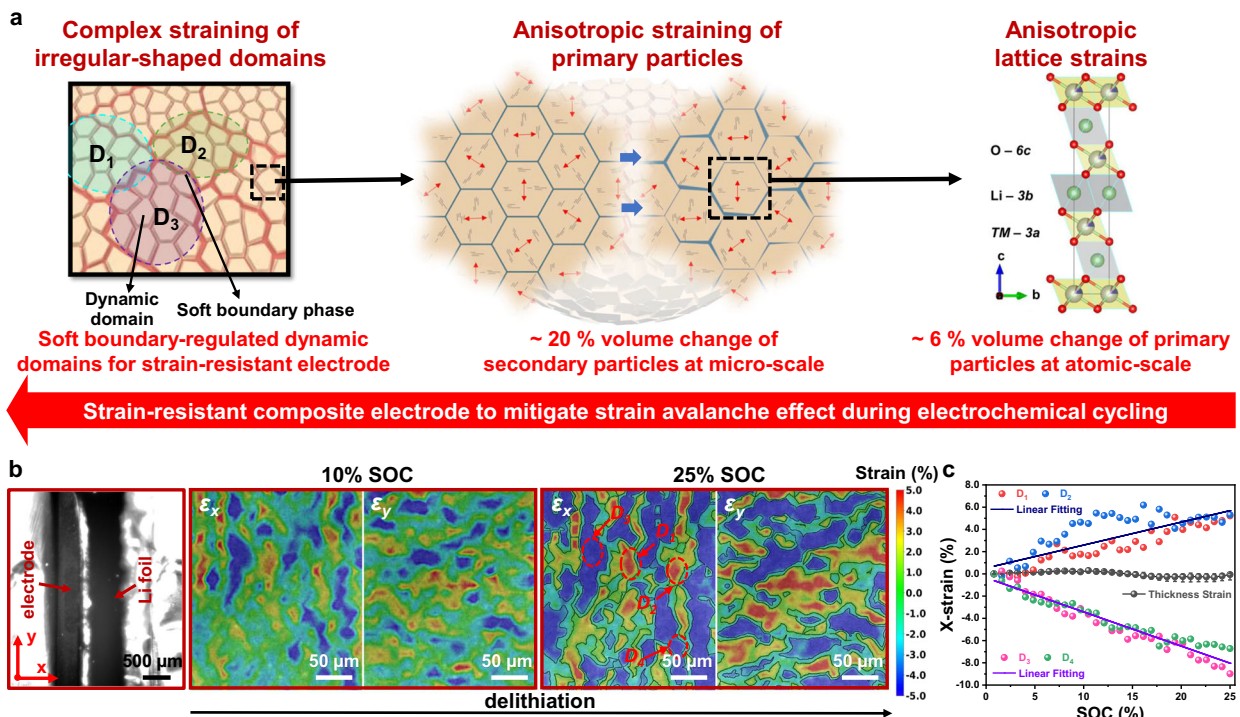

**Fig. 4 | Operando DIC characterizations of electrochemically induced straining.** **a** Schematic illustration of the strain avalanche effect, which originates at the atomic-scale lattice, propagates through micro-scale secondary particles, and extends to mesoscale domains. **b** Operando x- and y-strain mappings (via DIC analysis) showing the cross-sectional view of the composite electrode (360 μm thick) in the same region at different SOCs during delithiation at 2.0 mA cm$^{-2}$ under a voltage window of 2.7–4.3 V vs. Li| Li$^+$ without applying the external force. **c** Electrochemical-induced strains of the total electrode thickness and representative domains at various SOCs. Source data are provided as a Source Data file.

through the densely packed NMC811 particles. The porosity was calculated to be 8.42% based on the segmented volume, which is lower than the theoretical estimate of 12.4% through a rough calculation (Supplementary Text). Additionally, the volume fraction of the conductive PILG phase was independently estimated via deconvolution of TGA data, yielding a value of 30.22 vol%, which is in close agreement with the FIB-SEM-derived value of 34.72 vol% (including porosity) (Supplementary Table 5). This consistency validates the accuracy and reliability of our design and characterization methodology.

### Boundary phase-regulated strain-resistant composite electrodes for mitigating mechanochemical degradations during battery cycling

In battery applications, the NMC811 secondary particles, which are composed of primary single-crystal particles, undergo cyclic electrochemical loading due to repeated Li insertion and removal. This cyclic process induces a strain response similar to that caused by external dynamic mechanical forces. Of particular concern is the extreme anisotropic straining of primary particles (e.g., approximately 6% volumetric strain with 2% *c*-lattice expansion and 2% *a*-lattice contraction) during the delithiation process, which results in amplified strains within NMC811 secondary particles (e.g., 20% volume expansion) (Fig. 4a)[15,35–37]. This strain avalanche effect leads to mechanochemical degradation, negatively impacting battery performance. The issue is further exacerbated in thicker, denser electrodes, where complex strain fields originating at randomly distributed secondary particles propagate throughout the electrode. Our damage-tolerant composites mitigate these dynamic mechanical strains, preserving the integrity of the boundary phase and maintaining its connectivity with active secondary particles and their integrated domains during cycling.

To visualize the strain responses induced by electrochemical cycling in our densified NMC811-PILG composite electrode, we performed operando DIC to capture full-field dynamic strain mapping. The positive electrode, separator, and Li foil were assembled into a split cell without applying external force for the operando DIC analysis. The x- and y-strain mappings from a cross-sectional view of our electrode were analyzed during delithiation at a current density of 2.0 mA cm$^{-2}$ (Fig. 4b). A range of local strains, from positive (red-colored regions) to negative (blue-colored regions), was observed across different domains. For example, along the thickness direction, positive 5% x-strains ($\varepsilon_x$) at 25% state of charge (SOC) were observed in representative domains $D_1$ and $D_2$, while negative 7–9% x-strains were noted in local domains $D_3$ and $D_4$ (Fig. 4b, c). Interestingly, nearly zero net strain was observed along the overall electrode thickness (e.g., -0–0.3% x-strain at 0–25% SOCs, Fig. 4c). These distinct local strains are likely caused by the electrochemical-induced anisotropic deformations of irregular-shaped domains. It appears that the positive local strains in some domains are offset by the negative strains in neighboring domains. The surrounding soft boundaries may accommodate the complementary deformations of these dynamic domains, resulting in a globally strain-resistant feature throughout electrodes for stable cycling performance.

In contrast to the relatively large strains observed in the local active domains, the green-colored regions between them exhibit 0% x- or y-strains, indicating stable dynamic interfaces across the electrode during the delithiation process. By extracting these interface profiles from both x- and y-strain mappings and overlaying them into a comprehensive profile map, we identified common 0% strained sites, highlighted by red spots in Supplementary Fig. 15. This analysis reveals that the complex straining of irregular-shaped domains, driven by the strain avalanche effect from atomic lattices to primary and secondary active particles, is well constrained within our densified thick electrode. Moreover, the conducting secondary boundary phase, which adheres effectively to the dynamic active domains, enhances electro-

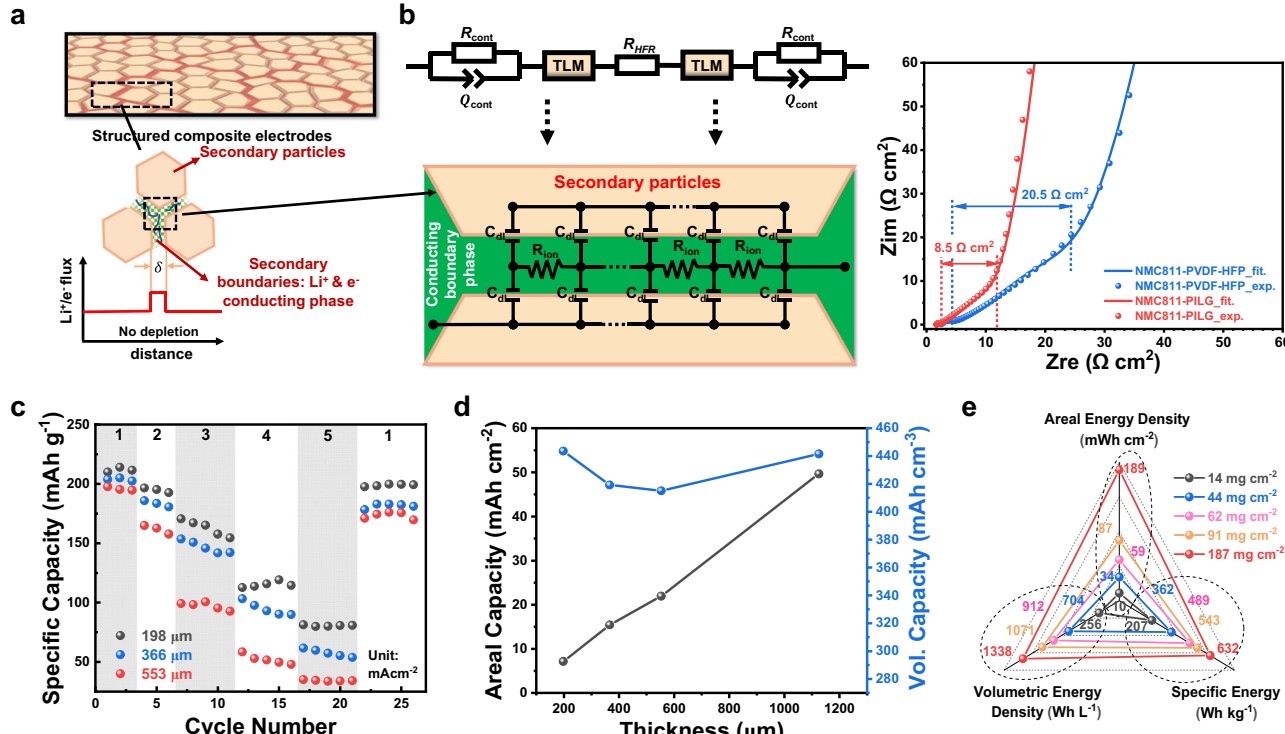

**Fig. 5 | Electrochemical characterizations of densified composite electrodes with varying thicknesses. a** Schematic illustration of conducting secondary boundary phases in densified electrodes, analogous to irrigation nourishing a dry landscape; **b** Transmission line model used to investigate charge transport via potentiostatic EIS in symmetric cells (NMC811-PILG ||NMC811-PILG) at 0% SOC. **c** Rate performance of NMC811(73.9%)-PILG positive electrodes with thicknesses of 198, 366, and 553 μm tested under a voltage window of 2.7–4.3 V vs. Li| Li⁺. **d** Thickness-dependent areal and volumetric capacities of NMC811(73.9%)-PILG positive electrodes at a current density of 1 mA cm⁻². The 1125-μm-thick electrode was pre-soaked in liquid electrolyte before testing. **e** Comparison of comprehensive cell-level performance, including areal energy density, volumetric energy density, and specific energy, between our densified electrodes (mass loading: 44–187 mg cm⁻²) and a slurry-coated electrode with a practical mass loading of 14 mg cm⁻². Cell-level energy density calculations are provided in the Supplementary Text. Source data are provided as a Source Data file.

chemo-mechanical coupling, as demonstrated by the pronounced strain evolution across various SOC levels from 0% to 25% during battery cycling. This strong coupling ensures highly accessible capacity throughout the entire densified thick electrode (360 μm) at a current density of 2 mA cm⁻². Additionally, we applied an external mechanical force to the split cell to dramatically reduce contact resistance between gold-coated stainless steel current collectors and electrodes during battery cycling (Supplementary Fig. 16). It demonstrated robust electro-chemo-mechanical coupling over an extended SOC range from 0% to 100% during the delithiation process at a current density of 1 mA cm⁻². The observed heterogeneous strains in local domains were well constrained within our strain-resistant composite electrode, as evidenced by minimized overall strains along both thickness and in-plane directions of the thick electrode.

### Conductive secondary boundary phase for tailored electrochemical performance

The engineered boundary phase not only enhances mechanical robustness but also serves as a "reservoir" for both ions and electrons, preventing charge depletion across interfacial boundaries. These secondary boundaries provide interconnected 3D charge transport pathways, much like how irrigation nourishes a dry landscape (Fig. 5a). To further study charge transport kinetics, electrochemical impedance spectroscopy (EIS) was conducted in a symmetric cell configuration using two identical positive electrodes (NMC811-PILG||NMC811-PILG)[38]. Validated by a transmission line model (TLM) (Fig. 5b and Supplementary Text), the Nyquist plots exhibited a 45° slope between ~5 and 100 Hz and quasi-vertical lines at lower frequencies (<1 Hz) during a nonfaradaic process at 0% SOC. The projection of the 45° slope onto

the real axis reflects the ionic resistance ($R_{ion}$/3) of the conducting boundary phase[39], which limits battery performance as electrodes become thicker and denser[40,41]. Consequently, the NMC811-PILG positive electrode, with a locally Li⁺-enriched gel electrolyte in secondary boundaries, showed much lower ionic resistance compared to the NMC811-PVDF-HFP electrode (e.g., 8.5 Ω·cm² vs. 20.5 Ω·cm², Fig. 5b).

Lithium-ion diffusivity, characterized using the galvanostatic intermittent titration technique (GITT) in a half cell, further confirmed the enhanced ion transport within the NMC811-PILG composite electrode. The calculated diffusion coefficient exceeds $2 \times 10^{-9}$ cm² s⁻¹ at a SOC ranging from 10% to 50%, surpassing that of the NMC811-PVDF-HFP electrode (Supplementary Fig. 17, associated equations and discussions in Supplementary Text). As a result, the NMC811-PILG electrode delivered higher capacities than the NMC811-PVDF-HFP electrode at current densities ranging from 1 to 5 mA cm⁻² (Supplementary Fig. 18). At higher current densities, the thick NMC811-PVDF-HFP electrode exhibited a more significant capacity reduction, primarily due to a large Li⁺ concentration gradient within the liquid electrolyte-filled electrode, which impedes efficient Li⁺ transports[42]. In contrast, the NMC811-PILG electrode, integrated with the Li⁺-enriched PILG conducting phase, maintained local oversaturation of Li⁺, preventing depletion during lithium insertion (i.e., discharge).

To assess the impact of electrode thickness on charge transport kinetics of conducting secondary boundary phase, we varied the thickness of NMC811 (73.9%)-PILG composite electrodes from 180 to 534 μm. The Nyquist plots revealed a quasi-linear correlation between ionic resistance ($R_{ion}$) and electrode thickness, demonstrating nearly constant resistivity (Supplementary Figs. 19, 20, and Table 6). Typically, porous thick electrodes filled with liquid electrolytes suffer from

non-uniform charge distribution, leading to local SOC heterogeneities and severely degraded battery performance. However, our transient liquid-assisted densification process, incorporating a $Li^+$-enriched conducting secondary boundary phase evenly distributed around active materials, ensured sufficient $Li^+$ supply and robust mechanical integrity to withstand dynamic local strains during battery cycling[15]. Consequently, NMC811-PILG electrodes with various thicknesses achieved nearly theoretical specific capacities of 195–212 mAh g$^{-1}$ at a current density of 1 mA cm$^{-2}$ (Fig. 5c and Supplementary Fig. 21), indicating almost full utilization of active materials. This performance exceeds the critical thickness threshold of 200 μm identified in previous studies[43], which linked increased electrode thickness to rapid capacity loss and underutilization of active materials due to diffusion limitations and $Li^+$ depletion. Furthermore, as the current density increased from 1 to 5 mA cm$^{-2}$, differences in specific capacity for various electrode thicknesses became more pronounced due to thickness-dependent ionic resistance (Fig. 5c). When the current density returned to 1 mA cm$^{-2}$, these thick electrodes restored most of their capacities, indicating the robustness of our composite electrodes during repeated charge-discharge cycles. Our thick electrodes were also characterized by in situ EIS and cyclic voltammetry (CV) analysis (Supplementary Figs. 22, 23).

To evaluate the potential high-energy-density applications, both areal and volumetric capacities were measured as the thickness increased. A linear increase in areal capacity up to 50 mAh cm$^{-2}$ was observed at a current density of 1 mA cm$^{-2}$ as the thickness increased from around 200 to over 1100 μm (Fig. 5d). These electrodes delivered high volumetric capacities of 415–445 mAh cm$^{-3}$ regardless of thickness, enabled by the rapid charge transport kinetics provided by the conducting boundary phase within the densified electrodes (porosity: 8–15%).

Building on these results, we further benchmarked the comprehensive cell-level performance of our densified thick electrodes, which spanned a wide mass loading range of 44–187 mg cm$^{-2}$, against a conventional slurry-coated electrode with a practical mass loading of 14 mg cm$^{-2}$ (Fig. 5e, see Supplementary Text for the calculation of cell-level energy density). The cell-level specific (362–632 Wh kg$^{-1}$), volumetric (704–1338 Wh L$^{-1}$), and areal energy densities (34–189 mWh cm$^{-2}$) of our densified thick electrodes significantly outperformed those of the slurry-coated electrode (207 Wh kg$^{-1}$, 256 Wh L$^{-1}$, and 10 mWh cm$^{-2}$). These improvements are attributed to the sustained charge transport kinetics enabled by the conductive boundary phases, along with simultaneously increased electrode density and mass loading, collectively delivering enhanced electrochemical performance across all key metrics.

## Enhanced electrochemical performance with increased active material content

The good electrochemical performance presented earlier was achieved using densified thick composite electrodes with a relatively low active material percentage of 73.9 wt%. To further enhance practical battery performance, increasing the active material content while preserving efficient charge transport within the composite architecture is essential. As shown in Fig. 3, the conducting boundary phase remains uniformly distributed across composites with varying NMC811 contents of 73.9 wt%, 86.7 wt%, and 92.7 wt%, suggesting structural adaptability of composite electrodes with integrated charge transport networks.

To further assess the charge transport properties for both ions and electrons, we conducted EIS and direct current (DC) polarization measurements (Supplementary Fig. 24)[38]. Nyquist plots obtained from potentiostatic EIS revealed a clear trend of increasing ionic resistance ($R_{ion}$) with decreasing conducting PILG boundary phase (Supplementary Fig. 24a and Table 6), resulting in reduced rate capability at increasing current densities from 1 to 5 mA cm$^{-2}$ (Fig. 6a). To isolate

electronic conductivity, DC polarization tests were carried out by applying a small constant voltage. The initial current comprising both ionic and electronic components rapidly decays as ionic migration is suppressed, and the remaining steady-state current predominantly reflects electronic transport through the composite (Supplementary Fig. 24b)[44]. Notably, the NMC811 (92.7%)–PILG positive electrode exhibited the highest steady-state current despite containing the lowest fraction of carbon additives. This enhanced electronic conductivity is attributed to an optimized composite microstructure, in which the reduced PILG fraction concentrates the electronically conductive carbon network (CNF and graphene), increasing the effective ratio of electron-conductive to ion-conductive components (e.g., CNF +graphene/PILG = 0.97:1 for NMC811 (92.7%)–PILG, compared to 0.74:1 for NMC811 (73.9%)–PILG and 0.77:1 for NMC811 (86.7%)–PILG, according to Supplementary Table 3). This efficient percolated carbon network facilitates superior electron transport even at high active material contents.

At a rate of 0.05 C, the NMC811(73.9%)–PILG positive electrode delivered a specific capacity of 218.5 mAh g$^{-1}$ with an initial coulombic efficiency (ICE) of 88.82%; the NMC811(86.7%)–PILG electrode exhibited 197.9 mAh g$^{-1}$ with an ICE of 88.84%; and the NMC811(92.7%)–PILG electrode achieved 204.5 mAh g$^{-1}$ with a higher ICE of 91.67% (Fig. 6a). Among the three, the electrode with the lowest active material content (73.9%) demonstrated the highest specific capacities across various current densities (1–5 mA cm$^{-2}$), attributed to its lower ionic resistance compared to electrodes with higher active material contents (Fig. 6a). Importantly, all thick electrodes recovered most of their capacity when the current density was returned to 1 mA cm$^{-2}$, highlighting good rate reversibility.

Despite its relatively lower specific capacity, the NMC811(92.7%)–PILG electrode delivered the highest volumetric capacity, approaching 500 mAh cm$^{-3}$ during the first 10 cycles, owing to its elevated active material content. However, it exhibited reduced capacity retention (e.g., 67.7%) compared to the electrodes with 73.9% (e.g., 83.0%) and 86.7% (e.g., 77.6%) active material contents upon 100 cycling, highlighting an inherent trade-off between gravimetric and volumetric performance (Fig. 6b). These findings emphasize the critical need to optimize charge transport kinetics within thick and dense electrodes in order to simultaneously enhance gravimetric and volumetric performance.

A balanced electrode with an intermediate active material content of 86.7% was selected to evaluate long-term cycling performance. It retained a capacity above 121.8 mAh g$^{-1}$ after 300 cycles, corresponding to a capacity retention of 65.4% (Fig. 6c). Although its capacity retention remains lower than that of slurry-coated thin-film electrodes, the result is promising for structured electrodes designed with increased density (porosity <8–15%) and thickness (>200 μm), thanks to the improved charge transport enabled by synthetic conducting boundary phases.

Post-cycling SEM–EDS analysis was conducted to examine the structural integrity of the composite electrodes. Despite some increase in surface porosity (Top-view SEM image in Fig. 6d), the electrodes maintained good structural integrity after extended cycling, suggesting that the PILG phase effectively mitigates mechanical degradation[36]. Cross-sectional SEM images revealed that most NMC811 secondary particles retained their morphology, though some pulverization was observed near the separator-facing surface (Fig. 6d). Additionally, EDS line scans showed elevated oxygen signals near the surface compared to pre-cycling conditions, while nickel signals remained largely unchanged (Supplementary Fig. 25). This suggests the possible formation of oxygen-rich species such as LiOH, $Li_2CO_3$, NiO, and others, likely due to cathode–electrolyte interphase (CEI) formation under prolonged cycling[45,46].

Although the optimized architecture of positive electrodes largely suppresses electromechanical degradation, capacity fading can still

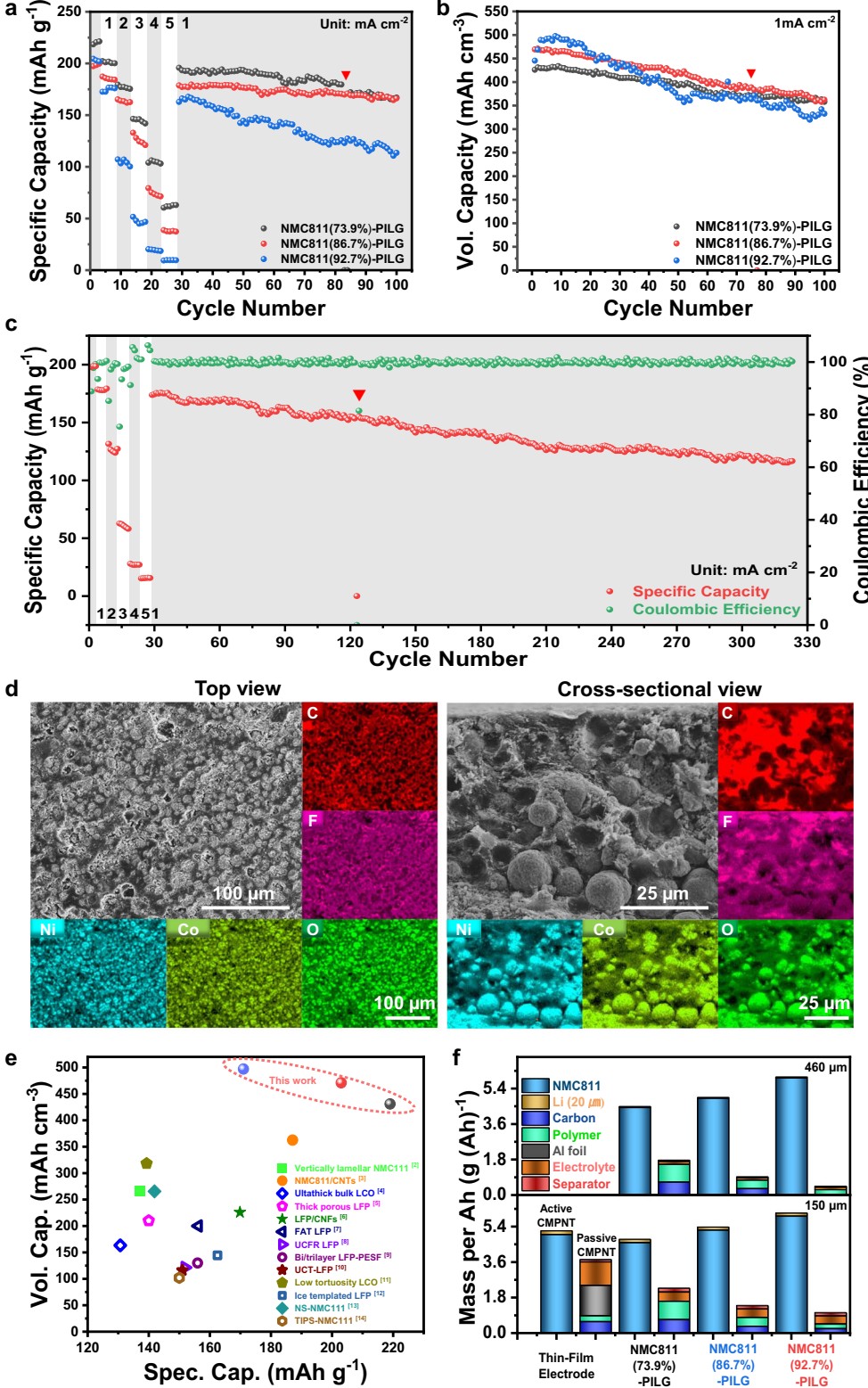

occur due to unstable lithium plating and stripping at the lithium metal negative electrode, which promotes dendrite formation (Supplementary Fig. 26). The capacity of cycled cells could be largely restored after replacing the lithium foil, as indicated by the red triangles in Fig. 6a–c. Addressing this persistent challenge through interfacial and structural engineering of the lithium metal negative electrode, or by incorporating solid-state electrolytes, could further improve cycling stability

and facilitate the practical implementation of our thick electrode design[47–49].

We compared the performance of our densified electrodes with previously reported architected thick electrodes (Fig. 6e and Supplementary Table 7)[2–14]. Our optimized positive electrodes are competitive in balancing gravimetric and volumetric capacity relative to representative reports in the literature. For example, at similar or even

**Fig. 6 | Electrochemical characterizations of densified composite electrodes with varying active material contents in Li ||NMC811·PILG cells under a voltage window of 2.7–4.3 V vs. Li| Li⁺. a** Specific capacities (normalized to active material) of NMC811·PILG electrodes with active material contents of 73.9%, 86.7%, and 92.7% (thickness: >150 μm, relative density: 86–88%). **b** Volumetric cycling performance (normalized to the total electrode) of various NMC811·PILG electrodes (thickness: >150 μm, relative density: 86–88%). **c** Long-term cycling of the NMC811 (86.7%)·PILG electrode (thickness of positive electrode: 215 μm, relative density: 86%). The current densities are indicated in (**a–c**). **d** Top and cross-sectional views of the cycled NMC811 (86.7%)·PILG electrode, along with corresponding elemental mapping of C, F, Ni, Co, and O obtained by EDS. **e** Comparison of volumetric and specific capacities of our densified thick positive electrodes with various architected thick electrodes reported in the literature[2–14]. Sources of literature data are listed in Supplementary Information, Table 7. **f** Comparison of the mass per ampere-hour of various active and passive components between slurry-coated thin-film and our NMC811·PILG positive electrodes with different active material loadings (excluding lead and packaging materials; top panel: ~460 μm electrode thickness; bottom panel: ~150 μm electrode thickness). Red triangles indicate cycles in which a new Li foil was replaced (**a–c**). Source data are provided as a Source Data file.

higher specific capacities, our electrodes delivered much higher volumetric capacities than prior studies (430–500 mAh cm⁻³ vs. <100–350 mAh cm⁻³).

More importantly, our thick and dense electrodes offer distinct advantages over porous, thin-film electrodes for real-world applications. As the active material content increases, the mass of passive components required per ampere-hour (Ah) is significantly reduced, thereby boosting cell-level energy density (Fig. 6f and Supplementary Tables 8–10). Meanwhile, the overall volume required per Ah also decreases, indicating strong potential for space-constrained applications (Supplementary Fig. 27 and Supplementary Tables 8–10)[50,51].

### Potential for scalable and efficient manufacturing
Our transient liquid-assisted densification process offers a highly energy-efficient route for densifying composites with high inorganic content. Based on an empirical heating profile, we estimate that this process consumes two to three orders of magnitude less energy than spark plasma sintering (SPS), one of the most energy-efficient sintering techniques currently available, as well as conventional high-temperature sintering methods (Supplementary Fig. 28)[5,41,52–58].

Scaling up this densification strategy is crucial for enabling practical, high-throughput production. As a proof of concept, we successfully fabricated a large-format electrode using a custom rectangular die with dimensions of 57 mm × 46 mm, demonstrating the feasibility of adapting this method for future pouch-cell applications. Production throughput can be further improved by implementing a serial pressing configuration, which allows the simultaneous densification of multiple samples[59]. Beyond batch-scale processing, our approach could be adapted to a continuous pilot-scale manufacturing line using hot roller pressing, a mature and industrially scalable metal forming technique. In this envisioned setup, a sequence of pressure- and temperature-controlled rollers would be integrated into a roll-to-roll production system, supported by in-line, non-destructive quality control techniques. Despite its promising potential, the widespread commercialization of this fabrication method will require overcoming several scientific and engineering challenges. These include: ensuring uniform dispersion of the transient liquid phase, achieving consistent particle rearrangement and compaction, minimizing stress gradients under uniaxial pressure, promoting homogeneous heat transfer, and enabling uniform removal of transient liquids during processing.

### Discussion
This work demonstrates that transient liquids can effectively drive pressure solution creep to enable low-temperature densification of composite electrodes at just 120 °C. The process presents an energy-efficient and versatile alternative to traditional sintering routes for inorganic powder manufacturing. Central to this approach is the formation of a synthetic conducting boundary phase via transient liquid mediation, which enhances the structural integrity of the composite electrodes and mitigates mechanochemical degradation during cycling. Furthermore, operando DIC reveals full-field strain evolution within thick and dense electrodes, providing critical insight into their strain-resilient behavior. The conducting secondary boundary phase also facilitates efficient charge transport throughout the dense and thick architecture, enabling enhanced electrochemical performance across all key metrics (including gravimetric, areal, and volumetric performance) that exceeds both conventional slurry-coated thin-film electrodes and state-of-the-art architected thick electrodes. Altogether, our strategy supports the design and fabrication of high-performance composite electrodes tailored for demanding energy storage applications.

## Methods
### Materials
The commercial NMC811 powders and Li metal chip (purity >99.9%, thickness: 0.6 mm, diameter: 16 mm, density: 0.534 g/cm³) were purchased from MTI Co., Ltd. Both materials were opened and stored in an Ar-filled glove box ($H_2O < 0.1$ ppm, $O_2 < 1.0$ ppm). Celgard 2340 separator (thickness: 38 μm, porosity: 45%), consisting of a polypropylene(PP)/Polyethylene(PE)/PP trilayer membrane, was purchased from Celgard. Carbon nanofiber (CNF), Poly (vinylidene fluoride-co-hexafluoropropylene) (PVDF-HFP), 1-Ethyl-3-methylimidazolium bis(trifluoromethylsulfonyl)imide (EMIMTFSI) IL ( >99%, $H_2O < 500$ ppm), Dimethylformamide (DMF), Anhydrous acetone ($H_2O < 0.01$%), Bis(trifluoromethane)sulfonimide lithium salt (LiTFSI) and 1.0 M Lithium hexafluorophosphate ($LiPF_6$) in a 1:1:1 (v/v/v) mixture of ethylene carbonate (EC), dimethyl carbonate (DMC), and diethylene carbonate (DEC), were purchased from Sigma-Aldrich. Graphene was purchased from MSE Supplies LLC.

### Preparation of PVDF-HFP-based polymer stock solution
The PVDF-HFP acetone solution was prepared by dissolving 0.674 g PVDF-HFP in 7 mL of acetone and stirring at 50 °C for 5 h. The PVDF-HFP-based PILG stock solution was prepared by dissolving PVDF-HFP, IL, and LITFSI in 7 mL of acetone with a specific ratio (Supplementary Table 1) and stirring at 50 °C for 5 h.

### Fabrication of NMC composite electrodes with various liquids
Two batches of mixtures consisting of NMC811 powders, carbon nanofibers, and graphene nanosheets were dehydrated under vacuum at 120 °C, then well-mixed with the PILG stock solution using a Thinky centrifuge mixer. The first batch (~84 wt% NMC811, 10 wt% CNF-graphene additives) was further mixed with the PVDF-HFP acetone solution using a mortar and pestle to obtain the mixture with (w./) acetone, without (w./o.) IL. Half of this mixture was vacuum dried for 24 h to fully evaporate the acetone solvent, resulting in a mixture without (w./o.) acetone and IL. The second batch was mixed with the PILG acetone solution, resulting in a mixture with (w./) acetone (<3 wt%) and IL. Three sets of powders with different liquids were then mixed with (w./) or without (w./o.) a small amount of DMF solvent (10 μL/100 mg) using mortar grinding. Subsequently, the precursor was transferred into an Ar-filled glove box ($H_2O < 0.1$ ppm, $O_2 < 1.0$ ppm) and compressed into pellet form using a hydraulic press to fabricate the positive electrode. The quasi-solid mixture was transferred into a 10- or 13-mm diameter die and pressed under uniaxial pressure of 400 MPa at 25 °C. The die was then heated at 120 °C for 60 min at a heating rate of 10 °C min⁻¹. Six electrode pellets were prepared from

the process with or without acetone, DMF, and IL, respectively. During the densification process, acetone and DMF transient liquids evaporated, along with some IL loss out of the die. The weight ratios of the different components prior to the densification process are summarized in Supplementary Table 1 for composite electrodes containing 73.9%, 86.7%, and 92.7% active material.

## Material characterizations

Morphological and structural characteristics were analyzed using scanning electron microscopy (SEM, Apreo 5) equipped with EDS analysis (Oxford Instruments), XRD (Panalytical X'Pert Pro X-ray Powder Diffractometer), and FT-IR. FIB-SEM imaging was conducted using a Thermo Fisher Scientific Scios2 system to analyze the NMC811-PILG thick electrode. Gallium ($Ga^+$) ions were used for serial sectioning at a slice thickness of 50 nm. The process was automated with the Auto Slice & View 4 software, and subsequent image analysis was performed in Dragonfly. The 3D reconstructed image reveals three primary components of interest: NMC secondary particles, the PILG phase containing carbon additives, and pores. These components were segmented based on their relative contrast and intensity, following careful denoising. Thermogravimetric analysis (TGA, PerkinElmer instruments Pyris Diamond TG/DTA) was conducted in airflow from 25 °C to 800 °C at a heating rate of 10 °C min$^{-1}$. The pellet electrodes prepared with different transient liquids were cut into ~ 3 × 12 mm rectangular bars and loaded onto a dynamic mechanical analyzer (DMA) (Anton Paar MCR 702e MultiDrive) to measure tensile properties at a displacement rate of 1 mm min$^{-1}$. The shaped rectangular samples were also loaded onto a universal mechanical tester (CS2-1100 Series, Chatillon) equipped with a DIC setup to capture the full-field displacements/strains during the test at a displacement rate of 0.05 mm min$^{-1}$. Specifically, in situ DIC analysis was performed using GOM Correlate software (Trilion, PA, USA). A 12-MP 2D camera with a minimum spatial resolution of 25 μm (i.e., the smallest gauge area over which strain can be computed) was positioned in front of the composite specimens. Real-time, full-field displacements and strains were derived from time-resolved 2D grayscale images, enabling detailed dynamic mechanical analysis at specific sites[60]. The image of the undeformed specimen served as the reference. Viscosity was measured at a shear rate $\dot{\gamma} = 0.12^{-1}$ upon heating from 30 to 110 °C with a ramping rate of 10 °C min$^{-1}$ using the Rheometer (MCR 702e MultiDrive) via a plate-plate measuring system.

## Electrochemical measurements

All electrodes were assembled into two-electrode CR 2032 coin-type cells (Li ||NMC811-PILG), using lithium foil as the counter electrode, a Celgard 2340 membrane separator (punched to the appropriate size), and 100 μL of 1 M $LiPF_6$ in EC/DEC/DMC (1:1:1, v/v/v) as the electrolyte, dispensed with PP pipette tips. Cell assembly was performed in an argon-filled glove box with moisture and oxygen levels maintained below 0.1 ppm and 1.0 ppm, respectively. Galvanostatic charge-discharge cycles were tested using Landt Instruments at various current densities of 1–5 mA cm$^{-2}$, and a voltage window of 2.7–4.3 V vs. Li| Li$^+$. The Coulombic Efficiency (CE) was calculated as the ratio of the discharge capacity to the charge capacity in the preceding cycle. The specific, areal, and volumetric capacities of the positive electrode were calculated based on the weight of the active material, the total electrode area, and the total electrode volume, respectively. In this study, at least five cells were fabricated under each condition. The data presented in the figures are representative of the overall results and serve to demonstrate the reproducibility and reliability of the electrochemical performance. Potentiostatic EIS measurements (Ametek, Princeton Applied Research, Versa STAT 4) were performed at open-circuit potential using a sinusoidal AC perturbation with an amplitude of 10 mV, over the frequency range from 100 kHz down to 10 mHz, with 10 points per decade. Two identical positive electrodes were assembled into a symmetric cell (NMC811-PILG ||NMC811-PILG) for EIS measurements to determine ionic resistances. All impedance obtained from EIS measurements is multiplied by the geometrical areas of the electrodes. Nyquist plots were fitted using the EIS Spectrum Analyzer (EISSA) software. In situ EIS measurements at various SOCs were conducted every 1 h during the charging or discharging process at an areal current density of 1.0 mA cm$^{-2}$. Cyclic voltammetry (CV, Ametek, Princeton Applied Research, Versa STAT 4) was carried out at a scan rate of 0.5 mV s$^{-1}$ under a voltage window of 2.7–4.3 V vs. Li| Li$^+$. DC polarization curves of each composite positive electrode were measured using electron non-blocking electrodes (SUS|| NMC811-PILG|| SUS; SUS: stainless steel). The time-dependent electrical current was recorded under a constant voltage for 3600 s.

## Operando DIC characterization

The thick composite electrodes were cut to expose the fresh surface in the glove box ($O_2$ and $H_2O$ < 1.0 ppm). The sectioned positive electrode was vertically inserted into the in situ optical cell to serve as the working electrode for cross-sectional view observation. A same-sized Li metal plate was vertically inserted into the other side of the cell to serve as the counter electrode. Together with the working and counter electrodes, aluminum (Al) and copper (Cu) foils were attached to the back and extended to the outside of the cell with two threads of Cu wires, respectively. A glass fiber separator was sandwiched between the working and the counter electrodes and further saturated with a liquid electrolyte. The cell was sealed by the screw-fastened quartz window and surrounding O-rings. This cell setup was charged-discharged using a Landt cycler and captured with a DIC camera simultaneously for the full-field observation under the battery operation conditions (e.g., 1–2 mA cm$^{-2}$, a voltage window of 2.7–4.3 V vs. Li| Li$^+$). Moreover, these operando DIC measurements were performed both without external force (Fig. 4b) and under an applied external force (Supplementary Fig. 16).

## Data availability

The source data generated in this study are provided in the Source Data file. Source data are provided with this paper.

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

## Acknowledgements

H.S. acknowledges financial support from the Pennsylvania State University start-up fund and the National Science Foundation under Award CBET-2134643. S.L. thanks the Peter and Angela Dal Pezzo Graduate Fellowship.

## Author contributions

H.S. conceived the research. B.N. and S.L. conducted composite synthesis, material characterization, and electrochemical measurements. B.N., T.W., and T.L. contributed to DIC measurements and post analyses. H.S. and J.L. supervised the research. H.S., B.N., T.L., and S.L. co-wrote the manuscript with input from all the authors. All authors discussed the results and commented on the manuscript.

## Competing interests

H.S., B.N., S.L., and T.W. are inventors on patents (US Provisional Application No. 63/840,217 and US Provisional Application No. 63/840,233) relating to this study filed by the Pennsylvania State University, University Park. H.S., B.N., S.L., and T.W. have no other competing interests. The remaining authors declare no competing interests.
