## [Transparent Peer Review file · Nature Communications]

Unveiling Multifunctional Synthetic Boundaries for Enhanced Mechanical and Electrochemical Performance in Densified Thick Composite Electrodes

Corresponding Author: Professor Hongtao Sun

Version 0:

Reviewer comments:

Reviewer #1

(Remarks to the Author)

This manuscript explored a geology-inspired densification process to create dense and thick electrodes, which is crucial for high-energy-density lithium-ion batteries. The research claims that forming synthetic secondary boundaries within these electrodes provides strain-resistant properties, enhances charge transport, and facilitates direct recycling, positioning this as a method for advancing sustainable battery technologies. While the study emphasizes advancements through the development of synthetic secondary boundaries, the real-world application of such thick and dense electrodes remains questionable due to the scaling challenges and manufacturing costs associated with the technology. Moreover, the strategy in this work does not address the long-term operational stability and durability of thick electrodes. The conclusions drawn about the practicality of direct recycling and improved battery performance need more rigorous validation against industry standards. The methodology, while innovative, requires more detailed comparison with existing solutions to better quantify improvements and justify the additional complexity and potential cost. Therefore, this manuscript does not meet the publication requirements.

Here are more detailed evaluations and comments:

1. The author believes that "Capacity fading was primarily attributed to unstable Li plating/stripping and dendrite growth on the Li metal anode during cycling" However, based on the cycling performance demonstrated in Figure S18, the battery capacity continues to decline even after replacing the fresh lithium metal, indicating that the cathode faces severe degradation during cycling, rather than the lithium metal being the sole cause of capacity fade. Therefore, the thick electrode prepared by this method does not exhibit satisfactory stability, with significant degradation occurring within at least 100 cycles.
2. Essentially, the NMC-PILG electrode incorporates additional lithium salt during manufacturing compared to NMC-PVDF-HFP, which inherently enhances ionic conductivity. It is conceivable that the performance improvement may not solely attribute to the densification process via pressure-solution creep as described by the authors.
3. This work achieves direct recycling of failed electrodes by breaking and re-forming the synthetic boundaries within the electrode. There is no direct evidence suggesting that electrode failure originates from the failure of synthetic boundaries; rather, it is more likely due to structural degradation of the NMC particles themselves. During secondary utilization, relithiation occurs, and it is inevitable that the electrode capacity will increase again after relithiation, not necessarily due to the regeneration of synthetic boundaries.
4. While the approach to using geology-inspired densification is novel, its transformative impact on the battery manufacturing industry remains speculative. The innovation is well-noted but lacks a comparative analysis with other modern technologies that might offer similar benefits without the need for entirely new processes.
5. The manuscript concentrates on a synthetic boundary phase for thick electrodes, but the scope is limited, ignoring broader battery technologies such as solid-state or flexible batteries. This narrow focus may restrict the generalizability and impact of the findings.

6. Although the authors noticed the energy consumption of different manufacturing process, it is not the same with cost. It is necessary to discuss the cost implications of the novel manufacturing process, particularly in terms of upscaling and integration with current production lines, which is critical for commercial adoption.

7. Although this research has improved the performance of batteries, the environmental impact of organic solvents used in the production process should be considered, especially when large amounts of solvents are volatilized during mass production.

Reviewer #2

(Remarks to the Author)

This manuscript researches the damage tolerance of dense composite electrode is enhanced by the secondary boundary phase regulated by transient liquid, which solves the problem of mechanochemical degradation during the battery cycle, and this research work is interesting. However, the manuscript still existed some minor questions, and it should be undergone minor revisions before its acceptance by Nature Communications, as following:

1. All figures in the article should be annotated. For example: 12, 16 and 25 in Fig. 1a.

2. The article proposes that "It enhances damage-tolerance of composite electrodes, mitigating mechanochemical degradations during battery cycling;". In order to better demonstrate this view, comparative diagrams, data and explanations of evidence of pre - and post-cycle mechanochemical degradation should be provided as detailed as possible in the validation section.

3. Figure 5c should provide a corresponding equivalent circuit diagram.

4. If Figure 5j cycles is not illustrative, please provide a performance comparison chart with a larger number of cycles. Please provide a more detailed explanation and clarification on this manuscript.

Reviewer #3

(Remarks to the Author)

In this work, the author proposes a densification process for thick NCM811-PILG electrodes to achieve enhanced electrochemical performance. Starting with the description of the densification method, the author confirms the reduced porosity and the formation of synthetic secondary boundaries, which indicate improved electrode density. The analysis on reduced internal resistance, cycling performance under different current densities, and rate capability is conducted as well. The author concludes that the transient and ionic liquid-assisted densification process effectively enhances ion and electron transport pathways and enables high volumetric energy density. The author also provides a method of recycling active materials, giving more practicality and even sustainability.

The author develops the thick electrode with a conductive secondary boundary phase and thoroughly discusses the performance and applicability of the electrode. However, the reviewer believes that the claims could be reinforced with additional supporting data. The following are detailed comments:

1) The author has suggested that the decreased resistance observed in EIS data implies that the PILG effectively establishes ion and electron transfer pathways within the electrode. However, the data provided in Fig. 2 such as SEM images might not be a full representation to evaluate whether PILG has been uniformly deposited even at the pores located in depth. This raises the question of whether the observed improvement is due to uniform deposition or the sum of localized enhancements of properties. Could the author share any perspectives on this matter? Hence, the work could be reinforced if the author could provide any additional data to verify the uniformity of PILG distribution across the electrode pores?

2) Throughout this work, NCM811 is consistently utilized as the primary active material, likely chosen for its ability to achieve high energy density. However, the author employs NCM111 as the active material to evaluate recyclability. The reviewer is curious about the reasoning behind selecting NCM111 for this aspect of the study. Could the author elaborate on why NCM111 was deemed more suitable for recyclability testing? Furthermore, if the recycling of NCM811 is particularly challenging due to its structural instability, what alternative approaches or strategies could the author consider addressing this issue? More explanation in depth would further clarify the author's claims.

3) Continuing with the discussion on recycling experiment, the author provides the cell performance of recycled NCM111 in Fig. 5(j). The reviewer believes that the claims regarding the eco-friendliness of the designed electrodes could be reinforced by including a comparison between the cell performances of a newly fabricated electrode and a recycled electrode. The addition of this point could confirm the sustainable aspect of the proposed electrode.

4) The author states that the thick electrode with PILG exhibits the least swelling compared to its counterparts, indicating robust structural integration. The reviewer would appreciate further elaboration on any potential side reactions, the formation of a cathode electrolyte interphase (CEI), or pulverization, if observed. It would be helpful to obtain a more comprehensive view of the compatibility of this electrode with carbonate-based electrolytes.

Reviewer #4

(Remarks to the Author)

Manuscript No: NCOMMS-24-66206

Title: Unveiling multifunctional synthetic boundaries in densified thick composite electrodes to advance battery solutions

Decision: Reject

The manuscript reported liquid-polymer based densification method to fabricate thick NMC electrodes in the form of pellets. The impact of densification and its mechanical properties were studied followed by electrochemical performances. The objective of fabricating a robust thick battery electrode possessing a reasonable electrochemical performance is quite interesting. However, the manuscript lacks evidences to support the densification in terms of microstructural properties, cross-sectional analysis, ion trade-off at the interface, and prototype-level proof. In addition, the migration of polymer at such a high temperature is inevitable which raises serious concerns over the homogeneity of ion boundary or particle contact. Therefore, the manuscript is not suitable to be considered and I recommend to be rejected.

The comments are as follows,

1. The migration of polymer binder is inevitable under solution-processed electrodes obtained at 120 degrees. In such case, the homogeneity of ion-conductive boundary is not sufficiently validated.
2. Considering the mass loading of above 10 mg/cm² and content of NMC, the method would lead to severe migration/agglomeration of polymer.
3. The microstructural properties of densified cathodes, cycled and fresh ones via appropriate cross-section method is necessary. FIB-SEM could be such tool to reveal the microstructural properties.
4. The cross-section image depicted in Fig. 2b does not support the study.
5. The Li enriched boundaries across the cathode particles offer additional supply of Li-ions. The relative estimation of Li-ions that were available from electrolyte and poly(ionicliquid)-electrode is missing.
6. The authors mentioned a trade-off strategy, which must be accounted on the basis of ion-concentration in electrolyte and electrode.
7. The content of NMC811 in densified electrodes is low about 81% considering commercial NMC811 electrodes (wet-processed) about 94%. There is a large gap in the active content among the above conditions.
8. Porosity estimation through calculation is theoretical. Since, the dense or thick electrodes properties are dependent on the porosity, the authors must provide evidence based on tomography.
9. The tortuosity is an essential parameter for thick electrodes, the calculation of tortuosity for the thick NMC electrodes and related to performance should be provided.
10. Also, it is hard to compare the electrochemical data represented as mAh/cm³, mA/cm² with the previous reports. Suggest to report in a better format.
11. The prototype application is missing like full cell or against Li (Li metal battery) in a large format like punch cell.
12. There are some errors like LiNi_{0.80}Mn_{0.1}Co_{0.1}O₂ is NMC811 but not NCM811.

Version 1:

Reviewer comments:

Reviewer #1

(Remarks to the Author)

The authors have revised this work carefully. It can be published in Nature Commun now.

Reviewer #3

(Remarks to the Author)

The authors have thoroughly addressed all the concerns raised by the reviewer in the revised manuscript, and it is now recommended for publication.

Reviewer #4

(Remarks to the Author)

Manuscript ID: NCOMMS-24-066206

Title: Unveiling multifunctional synthetic boundaries for enhanced mechanical and electrochemical performance in densified thick composite electrode

Decision: Revise

Authors undertook the revision based on the comments raised and were addressed in the revised manuscript. However, there results related to electrode cross-section need more clarity.

1. The cross-section image of densified NMC cathodes is still not satisfactory. Suggest the authors to provide cross-sectional image of densified thick electrodes of various mass loading 44 mg/cm² to 187 mg/cm², which they have claimed.

Version 2:

Reviewer comments:

Reviewer #4

(Remarks to the Author)

Authors have performed relevant cross-section SEM of densified electrodes and were addressed in the revised manuscript. The current version could be accepted.

REVIEWER COMMENTS

Reviewer #1 (Remarks to the Author):

This manuscript explored a geology-inspired densification process to create dense and thick electrodes, which is crucial for high-energy-density lithium-ion batteries. The research claims that forming synthetic secondary boundaries within these electrodes provides strain-resistant properties, enhances charge transport, and facilitates direct recycling, positioning this as a method for advancing sustainable battery technologies. While the study emphasizes advancements through the development of synthetic secondary boundaries, the real-world application of such thick and dense electrodes remains questionable due to the scaling challenges and manufacturing costs associated with the technology. Moreover, the strategy in this work does not address the long-term operational stability and durability of thick electrodes. The conclusions drawn about the practicality of direct recycling and improved battery performance need more rigorous validation against industry standards. The methodology, while innovative, requires more detailed comparison with existing solutions to better quantify improvements and justify the additional complexity and potential cost. Therefore, this manuscript does not meet the publication requirements.

Response: We really appreciate the reviewer's constructive comments. The key innovation of our work is a geology-inspired densification process that creates synthetic secondary boundary phases within thick, dense composite electrodes. This design simultaneously delivers high gravimetric, areal, and volumetric performance—an outcome rarely achieved by existing methods (*e.g.*, slurry coating method). While increasing electrode thickness can improve device-level energy density, it often compromises gravimetric capacity through sluggish charge transport. Porous thick electrodes may deliver good gravimetric performance yet suffer from low volumetric energy density when porosity exceeds 40 percent. Our approach strikes an optimal balance, reducing porosity (*e.g.*, 8-15 %) while increasing thickness to preserve comprehensive electrochemical performance across all three metrics (gravimetric, areal, and volumetric performance).

We acknowledge that translating this novel processing method from lab scale (low technology readiness level) to industrial scale (high technology readiness level) involves significant scientific and engineering challenges. To date, we have consistently demonstrated the fabrication of high-performance, dense, and thick electrodes using a small die (12.7 mm diameter) for coin cell demonstrations. We have also initiated scale-up trials with a larger die (57 mm × 46 mm), achieving good electrode integration even as applied pressure decreases from 400 MPa to 50 MPa. To further address the scientific and engineering challenges beyond batch-level processing, we have assembled an interdisciplinary team of materials scientists, chemists, manufacturing engineers, and data scientists to develop a continuous pilot-scale roll-to-roll manufacturing process using hot roller pressing. This system incorporates multiple rollers with controlled pressure and temperature, as well as in-line, non-destructive monitoring, to ensure uniformity and quality. To further accelerate industrial translation, we will develop both physical and digital twins of the process, enabling closed-loop optimization and stringent quality control. We believe these sustained efforts underscore not only the scientific merit but also the potential practical viability of our densification strategy for next-generation battery electrodes in the near future.

We have removed the recycling studies and now focus exclusively on the core novelty of this work on the densification process, optimization of structured composite electrodes, and comprehensive

trade-off performance. In place of the removed content, we present additional validation efforts: (1) material and electrochemical characterization of composite electrodes with varying active-material contents of 73.9 wt%, 86.7 wt%, and 92.7 wt%; (2) projected performance analyses relevant to practical applications; and (3) 3D tomographic reconstructions to reveal internal structures and transport pathways. We believe these substantial revisions directly address the reviewer's concerns and significantly strengthen the manuscript for publication.

Here are more detailed evaluations and comments:

1. The author believes that “Capacity fading was primarily attributed to unstable Li plating/stripping and dendrite growth on the Li metal anode during cycling” However, based on the cycling performance demonstrated in Figure S18, the battery capacity continues to decline even after replacing the fresh lithium metal, indicating that the cathode faces severe degradation during cycling, rather than the lithium metal being the sole cause of capacity fade. Therefore, the thick electrode prepared by this method does not exhibit satisfactory stability, with significant degradation occurring within at least 100 cycles.

Response: We sincerely thank the reviewer for this important and insightful comment. As correctly noted, the capacity fading observed in our system cannot be solely attributed to lithium metal degradation. To prevent potential misinterpretation, we have revised the manuscript to clarify that the observed capacity decline results from a combination of factors, including degradation of the anode, cathode, and interfaces, as well as their chemical and electromechanical degradations.

In this revised manuscript, we first optimized the composition of electrically and ionically conductive components, along with the processing conditions, to increase the active material content from 73.9 wt% to 86.7 wt% and 92.7 wt%. Notably, increasing the active material content to 86.7 wt% not only enhanced the capacity normalized by total electrode, but also extended the cycle life to over 300 cycles (**Figure R1a–c**). Further increasing the content to 92.7 wt% led to an even higher volumetric capacity due to the increased active material content (*e.g.*, 500 mAh cm⁻³ for 92.7 wt% v.s. 475 mAh cm⁻³ for 86.7 wt% and 425 mAh cm⁻³ for 73.9 wt%); however, this came at the cost of reduced cycling stability.

To better understand the structural evolution during cycling, we performed post-cycling SEM-EDS analyses after 300 cycles. The results showed that the electrodes largely retained their structural integrity, with only limited pulverization localized near the separator-facing surface and a minor increase in surface porosity (**Figure R1d**). Furthermore, EDS line scans revealed elevated oxygen signals near the surface compared to the pre-cycling state, while nickel signals remained largely unchanged (**Figure R8c** or Supplementary Figure 24). These observations suggest the formation of oxygen-rich species—such as LiOH, Li₂CO₃, or NiO—likely associated with cathode–electrolyte interphase (CEI) formation during prolonged cycling.

Importantly, despite these localized degradation features, the overall electrode structure remained mechanically and chemically stable. We attribute this resilience to the presence of the 3D conductive PILG network, which plays a critical role in mitigating electromechanical degradation—a common challenge in conventional thick electrodes. This is further supported by cross-sectional thickness measurements, which indicate minimal swelling (only 1.04× to 1.08×) after 200–300 cycles.

In summary, through systematic optimization of the structure, composition, and active material content, we have demonstrated significantly extended cycle life in our dense and thick composite electrode system.

Figure R1. Electrochemical characterizations of densified composite electrodes with various active material contents. **a**, Rate performance of NMC811-PILG electrodes with various active material contents of 73.9 %, 86.7 %, and 92.7 % (thickness: $>150 \mu\text{m}$, relative density: 86-88 %). **b**, Cycling performance of various NMC811-PILG electrodes at a current density of 1 mA cm^{-2} (thickness: $>150 \mu\text{m}$, relative density: 86-88 %). **c**, Long-term cycling of the NMC811 (86.7 %)-PILG electrode (thickness: $215 \mu\text{m}$, relative density: 86%). **d**, Top view and cross-sectional view of the cycled NMC811 (86.7 %)-PILG electrode, along with corresponding elemental mapping of C, F, Ni, Co, and O obtained by EDS. Red triangles indicate the cycles in which a new Li foil was replaced (a, b, c).

2. Essentially, the NMC-PILG electrode incorporates additional lithium salt during manufacturing compared to NMC-PVDF-HFP, which inherently enhances ionic conductivity. It is conceivable that the performance improvement may not solely attribute to the densification process via pressure-solution creep as described by the authors.

Response: Thank you for the reviewer’s comments. We agree that Li-enriched conductive boundaries enhance charge-transport kinetics and improve battery performance. More importantly, our geology-inspired low temperature densification approach enables the seamless integration of diverse components into a highly compact form without thermal degradation. Therefore, it can incorporate various conducting components (electrically conductive carbon additives, ionically conductive PILG mixed phase) into synthetic conducting boundary phases that overcome transport limitations in thick, dense architectures. As a result, our composite electrodes—with areal mass loadings of 40 to 190 mg cm⁻², relative densities of 86 % to 88 %, and active-material contents of 73.9 % to 92.7 %—deliver simultaneously enhanced gravimetric (362–632 Wh kg⁻¹), areal (34–189 mWh cm⁻²), and volumetric (704–1338 Wh L⁻¹) performance that have not been achieved in previously reported electrodes (**Figure R2a**). In contrast, the slurry-coated electrode delivered only 207 Wh kg⁻¹, 256 Wh L⁻¹, and 10 mWh cm⁻². These improvements are attributed to the enhanced charge transport kinetics enabled by the conductive boundary phases, combined with increased electrode density and mass loading, which together deliver superior electrochemical performance across all three key metrics.

Furthermore, we compared our densified electrodes to previously reported architected thick electrodes, as shown in **Figure R2b** (updated Figure 6e and Supplementary Table 7 in the revised manuscript). Our optimized electrodes demonstrate a superior balance between gravimetric and volumetric capacity, outperforming literature values. For example, at comparable or even higher gravimetric capacities, our electrodes achieved substantially higher volumetric capacities than most prior studies (430–500 mAh cm⁻³ v.s. <100–350 mAh cm⁻³).

As part of our future investigation, we have optimized the composite electrodes to further increase the relative density from 86% to above 92%, thus delivering a high volumetric capacity of 607 mAh cm⁻³ at 0.05 C (**Figure R2c**) To the best of our knowledge, this represents one of the highest volumetric capacity reported. Our results emphasize that electrode performance is more strongly governed by microstructures and the conducting phase in our dense and thick NMC811–PILG composite system. Simply adding more lithium salts into the electrode system cannot achieve such promising trade-off electrochemical performance across all three key metrics. These investigations provide more insight into high-energy-density composite cathodes for the next-generation lithium batteries.

Figure R2. a, Comparison of comprehensive cell-level performance (including areal, volumetric, and gravimetric energy densities) between our densified electrodes (mass loading: 44-187 mg cm⁻²) and a slurry-coated electrode with a practical mass loading of 14 mg cm⁻². **b**, Comparison of volumetric and gravimetric capacities of our densified thick electrodes with various architected

thick electrodes reported in the literature. **c**, Cycling performance of the NMC811(86.7%)-PILG electrode (Thickness: 249 μm ; Relative density > 92 %)

3. This work achieves direct recycling of failed electrodes by breaking and re-forming the synthetic boundaries within the electrode. There is no direct evidence suggesting that electrode failure originates from the failure of synthetic boundaries; rather, it is more likely due to structural degradation of the NMC particles themselves. During secondary utilization, relithiation occurs, and it is inevitable that the electrode capacity will increase again after relithiation, not necessarily due to the regeneration of synthetic boundaries.

Response: We have removed the recycling studies and now focus exclusively on the core novelty of this work on the densification process, optimization of structured composite electrodes, and comprehensive trade-off performance. In place of the removed content, we present additional validation efforts: (1) material and electrochemical characterization of composite electrodes with varying active-material loadings; (2) projected performance analyses relevant to practical applications; and (3) 3D tomographic reconstructions to reveal internal structures and transport pathways (**Figures R10, 14** or Fig. 3k in the revised manuscript). We believe these substantial revisions significantly strengthen the manuscript for publication.

Direct recycling is another promising area that merits a separate, dedicated study. To clarify the role of the boundary phase in direct recycling, we have already obtained preliminary results that will be presented in a forthcoming manuscript. Specifically, we co-designed a synthetic boundary phase that enables both strong mechanical integration and rapid liquid-phase mass transport within the composite electrodes. This design allows for solution-based relithiation without breaking integrity of the composite electrodes. We have successfully demonstrated in situ relithiation of end-of-life composite electrodes using either a hydrothermal method at 120 °C or a room-temperature chemical process. Electrochemical characterizations of these one-step recycled electrodes show over 90% capacity recovery, highlighting the potential of this approach for sustainable battery reuse.

4. While the approach to using geology-inspired densification is novel, its transformative impact on the battery manufacturing industry remains speculative. The innovation is well-noted but lacks a comparative analysis with other modern technologies that might offer similar benefits without the need for entirely new processes.

Response: Thank you for raising this important point. Commercial battery electrodes are typically produced using a wet slurry-coating method, which presents several drawbacks: (1) it releases volatile and potentially toxic byproducts into the atmosphere; (2) it requires significant time and energy for electrode drying; and (3) it is generally limited to fabricating thin-film electrodes with low active material loadings, thereby constraining energy density when normalized by the total device weight.

In contrast, our approach utilizes only a minimal amount (1–5 vol%) of a transient solvent system—such as DMF or acetone—to initiate the localized dissolution of additives, including polymers and lithium salts. These additives then reprecipitate on pore surfaces during the thermomechanical process, promoting densification without using excessive fluids. This strategy enables the integration of dissimilar components and fabrication of thick and dense composite

electrodes that achieve significantly higher areal (*e.g.*, 7–50 mAh cm⁻²) and volumetric capacities (*e.g.*, 450–500 mAh cm⁻³) compared to those made with conventional techniques. Moreover, we anticipate that this densification method may also benefit all-solid-state batteries, a direction we have actively been exploring in our recent works.

Our method is still in the early stages of research and requires further demonstration for scale-up production and cell-level validation before it can move toward commercialization. In this work, we have demonstrated the feasibility of fabricating high-performance, dense, and thick electrodes using a small die (12.7 mm in diameter) for coin cells, showing consistent homogeneity across multiple batches. As part of our ongoing efforts, we are scaling up the process using a larger die for pouch-cell fabrication (**Figure R16**). The composite electrodes exhibit good structural integrity in a rectangular die (57 mm × 46 mm), even when the applied pressure is reduced from 400 MPa to 50 MPa. Battery performance using the pouch-cell configuration is currently under development. Beyond batch processing, we plan to develop a continuous pilot-scale manufacturing process using hot roller pressing—a well-established metal forming technique. This method will incorporate a series of rollers with controlled pressure and temperature, integrated into a continuous roll-to-roll production system. Various in-line, nondestructive monitoring techniques will be implemented to ensure quality control and structural consistency throughout the production process.

5. The manuscript concentrates on a synthetic boundary phase for thick electrodes, but the scope is limited, ignoring broader battery technologies such as solid-state or flexible batteries. This narrow focus may restrict the generalizability and impact of the findings.

Response: Our method offers significant advantages in integrating dissimilar materials into a highly compact form. In particular, it enables the densification of ceramics, metals, and hybrid organic–inorganic systems (*e.g.*, polymer-in-ceramic composites) at low temperatures. This capability is especially relevant for addressing solid–solid interfacial resistance, a key challenge in next-generation solid-state batteries. For instance, we have successfully fabricated solid-state electrolytes with a high inorganic content (*e.g.*, >85 wt%), including LATP–PILG polymer-in-ceramic systems and halide-in-ceramic composites (Figure R3, *Materials Today Energy* 2025, 49, 101829; *ACS Applied Materials & Interfaces*, 2024, 16, 67635). Conversely, by increasing the organic polymer content, the method can also be used to create polymer matrix composites for flexible electrodes and other battery devices.

Figure R3. Electrochemical performance of half-cell assembled with PILG phase-based oxide electrolytes (LATP-PILG). Rate performance and prolonged cycling at 0.1 C and 55 °C.

6. Although the authors noticed the energy consumption of different manufacturing process, it is not the same with cost. It is necessary to discuss the cost implications of the novel manufacturing process, particularly in terms of upscaling and integration with current production lines, which is critical for commercial adoption.

Response: We appreciate the reviewer's comment regarding the cost implications of our novel manufacturing process. We agree that a discussion on cost is essential for evaluating commercial viability. While our current batch-to-batch method using a small-sized die is not cost-efficient for large-scale manufacturing, our long-term objective is to transition to a continuous roll-to-roll production system. This will be achieved through hot roller pressing—a well-established and scalable metal-forming technique—which allows continuous densification with controlled pressure and temperature. In-line nondestructive monitoring techniques will be integrated to ensure consistent quality control, thereby improving process reliability and reducing waste. However, successful upscaling and integration with existing production lines will require overcoming several scientific and engineering challenges. These include ensuring even dispersion of the transient phase across particle surfaces, achieving uniform powder rearrangement and compaction, minimizing stress gradients during pressing, maintaining homogeneous heat transfer, and controlling the uniform evaporation of transient solvents. Addressing these challenges is part of our ongoing research and will be critical to making this low-temperature process both cost-effective and compatible with the current battery manufacturing infrastructure.

7. Although this research has improved the performance of batteries, the environmental impact of organic solvents used in the production process should be considered, especially when large amounts of solvents are volatilized during mass production.

Response: We thank the reviewer for raising this important point regarding the environmental implications of solvent use in our manufacturing process. We fully agree that solvent management is a critical consideration, especially for large-scale production.

Unlike conventional slurry-coating techniques that require large volumes of solvent, our method utilizes only a small amount (1–5 vol%) of a transient solvent system, such as DMF or acetone. This limited solvent is used to initiate localized dissolution of key additives (e.g., polymer binders and lithium salts), which subsequently re-precipitate onto pore surfaces during the thermomechanical densification process. This approach not only enhances densification at low processing temperatures, but also minimizes solvent volatilization, significantly reducing both the environmental footprint and energy consumption of the process.

In addition, our process strategically employs two organic solvents with distinctly different boiling points, enabling efficient separation and recovery via low-energy solvent recycling systems—technologies that are already widely used in scalable production environments. This solvent strategy allows for precise control over transient solvent evaporation during pressing and is compatible with industrial sustainability practices, facilitating easy integration into existing solvent recovery infrastructure with minimal energy input (see *Nature Reviews Clean Technology*, 2025, 1, 116–131; *Chinese Journal of Chemical Engineering*, 2025, 78, 273–283).

Furthermore, environmentally friendly, aqueous-based transient liquids are being developed for the fabrication of electrode active materials such as LiFePO_4 , which are inherently compatible with water-based systems.

Reviewer #2 (Remarks to the Author):

This manuscript researches the damage tolerance of dense composite electrode is enhanced by the secondary boundary phase regulated by transient liquid, which solves the problem of mechanochemical degradation during the battery cycle, and this research work is interesting. However, the manuscript still existed some minor questions, and it should be undergone minor revisions before its acceptance by Nature Communications, as following:

Response: We sincerely thank the reviewer for the positive evaluation of our work and for highlighting several important points. In response, we have carefully revised the manuscript to address all specific concerns. The key revisions include:

1. Adding complete annotations to all figures to enhance clarity and readability (e.g., updated Fig. 1a).
2. Incorporating comparative morphological and structural analyses before and after long-term cycling to more clearly demonstrate the mitigation of mechanochemical degradation.
3. Including an equivalent circuit diagram alongside the symmetric cell EIS data (now revised in Fig. 5) to support interpretation.
4. Expanding the performance comparison data to include extended cycling results, now accompanied by improved explanations and revised figures (e.g., updated Fig. 6).

We believe these comprehensive revisions significantly strengthen the manuscript by reinforcing our conclusions with additional experimental evidence and improving the overall clarity of presentation.

1. All figures in the article should be annotated. For example: 12, 16 and 25 in Fig. 1a.

Response: We thank the reviewer for bringing this to our attention. As suggested, all figures in the revised manuscript have been carefully reviewed and updated with appropriate annotations to ensure clarity and consistency, including the numerical labels in Fig. 1a.

2. The article proposes that “It enhances damage-tolerance of composite electrodes, mitigating mechanochemical degradations during battery cycling;”. In order to better demonstrate this view, comparative diagrams, data and explanations of evidence of pre - and post-cycle mechanochemical degradation should be provided as detailed as possible

Response: We appreciate the reviewer’s insightful comment. To address this point, we have included additional morphological characterizations using SEM-EDS analysis to investigate pre- and post-cycle mechanochemical degradation. Specifically, we examined three composite cathodes with different active material contents and performed SEM-EDS analyses before and after cycling (**Figure R4** or Figs. 3 and 6, Supplementary Fig. 24). The results showed that the NMC811 (86.7 %)-PILG electrode largely retained their structural integrity, with only limited pulverization localized near the separator-facing surface and a minor increase in surface porosity (**Figure R4d**). Furthermore, EDS line scans revealed elevated oxygen signals near the surface compared to the pre-cycling state, while nickel signals remained largely unchanged (**Figure R8c** or Supplementary Figure 24). These observations suggest the formation of oxygen-rich species—such as LiOH, Li₂CO₃, or NiO—likely associated with cathode–electrolyte interphase (CEI) formation during prolonged cycling.

Importantly, despite these localized degradation features, the overall electrode structure remained mechanically and chemically stable. We attribute this resilience to the presence of the 3D conductive PILG network, which plays a critical role in mitigating electromechanical degradation—a common challenge in conventional thick electrodes. This is further supported by cross-sectional thickness measurements, which indicate minimal thickness changes (only 1.04× to 1.08×) even after 200–300 cycles.

Figure R4 Morphology of pre- and post-cycled NMC811-PILG cathodes. **a-c**, Top view and cross-sectional view of densified composites, along with corresponding elemental mapping of C, F, Ni, Co, and O obtained by EDS for NMC811(86.7 %)-PILG, **d,e**, Top view and cross-sectional view of the cycled NMC811 (86.7 %)-PILG electrode, along with corresponding elemental mapping of C, F, Ni, Co, and O obtained by EDS. (**a-c**), Scale bars, 50 μm (**a**), 10 μm (**b**), 25 μm (**c** and **e**), 100 μm (**d**).

3. Figure 5c should provide a corresponding equivalent circuit diagram.

Response: Thank you for this helpful suggestion. The original symmetric cell EIS data, previously presented in Figure 5c, has now been relocated to Figure 5b in the revised manuscript. To facilitate clearer interpretation, we have added both an equivalent circuit diagram and a schematic illustration of the measurement setup in the new Figure 5a (also shown here as **Figure**

R5). These additions aim to provide a more intuitive and comprehensive understanding of the symmetric cell EIS results.

Figure R5. Electrochemical characterizations of densified composite electrodes a, Schematic illustration of conducting secondary boundary phases in densified electrodes, analogous to irrigation nourishing a dry landscape; **b,** Transmission line model and equivalent circuit used to investigate charge transport through the conducting boundary phase via potentiostatic EIS in symmetric cells.

4. If Figure 5j cycles is not illustrative, please provide a performance comparison chart with a larger number of cycles. Please provide a more detailed explanation and clarification on this manuscript.

Response: We sincerely thank the reviewer for raising this point. In the revised manuscript, we have removed the recycling studies and now focus exclusively on the core novelty of this work on the densification process, optimization of structured composite electrodes, and comprehensive trade-off performance. Moreover, we have incorporated additional experimental results comparing the cycling behaviors of cathodes with different active material contents as well as an extended cycling for optimized composite electrode as shown in **Figure R6** (or Fig. 6 in the revised manuscript).

Specifically, we optimized the composition and percentages of electrically and ionically conductive components, along with the processing conditions, to investigate thick composite electrodes with different active material contents of 73.9 wt% to 86.7 wt% and 92.7 wt%. Notably, among the three, the electrode with the lowest active material content (73.9 %) demonstrated the highest specific capacities across various current densities (1 to 5 mA cm⁻²), attributed to its lower ionic resistance compared to electrodes with higher active material contents (**Figure R6a**). Despite its relatively lower specific capacity, the NMC811(92.7%)–PILG electrode delivered the highest volumetric capacity, approaching 500 mAh cm⁻³ during the first 10 cycles, owing to its elevated active material content. However, it exhibited reduced capacity retention compared to the electrodes with 73.9 % and 86.7 % active material content, highlighting an inherent trade-off between gravimetric and volumetric performance (**Figure R6b**). These findings emphasize the critical need to optimize charge transport kinetics within thick and dense electrodes in order to simultaneously enhance gravimetric, areal, and volumetric performance metrics. A balanced electrode with an intermediate active material content of 86.7 % was selected to evaluate long-term cycling performance. It retained a capacity above 121.8 mAh g⁻¹ after 300 cycles (**Figure R6b**). Although its capacity retention remains lower than that of slurry-coated thin-film electrodes, the result is promising for structured electrodes designed with increased density (porosity < 8–

14 %) and thickness ($> 200 \mu\text{m}$), thanks to the improved charge transport enabled by synthetic conducting boundary phases.

Figure R6. Battery performance of densified composite electrodes with various active material contents. a, Rate performance of NMC811-PILG electrodes with various active material contents of 73.9 %, 86.7 %, and 92.7 % (thickness: $>150 \mu\text{m}$, relative density: 86-88 %). **b,** Cycling performance of various NMC811-PILG electrodes at a current density of 1 mA cm^{-2} (thickness: $>150 \mu\text{m}$, relative density: 86-88 %). **c,** Long-term cycling of the NMC811 (86.7 %)-PILG electrode (thickness: $215 \mu\text{m}$, relative density: 86%). Red triangles indicate the cycles in which a new Li foil was replaced.

Reviewer #3 (Remarks to the Author):
 In this work, the author proposes a densification process for thick NCM811-PILG electrodes to achieve enhanced electrochemical performance. Starting with the description of the densification method, the author confirms the reduced porosity and the formation of synthetic secondary boundaries, which indicate improved electrode density. The analysis on reduced internal resistance, cycling performance under different current densities, and rate capability is conducted as well. The author concludes that the transient and ionic liquid-assisted densification process effectively enhances ion and electron transport pathways and enables high volumetric energy density. The author also provides a method of recycling active materials, giving more practicality and even sustainability.

The author develops the thick electrode with a conductive secondary boundary phase and thoroughly discusses the performance and applicability of the electrode. However, the reviewer believes that the claims could be reinforced with additional supporting data. The following are detailed comments:

Response: We sincerely thank the reviewer for the thoughtful and constructive comments, as well as for the positive evaluation of our manuscript. We greatly appreciate the reviewer's encouragement to further strengthen our claims with additional experiments. In response, we have substantially revised the manuscript to include new experimental results and analyses that more clearly validate the structural and electrochemical advantages of our approach. These additions include:

- (1) Supplementary SEM and EDS characterizations before and after cycling across electrodes with different active material contents.
- (2) 3D reconstruction via FIB-SEM tomography and detailed post-analysis to elucidate the internal microstructure and phase distribution.
- (3) Extended cycling and rate performance data to demonstrate the long-term stability of our electrodes under practical conditions.
- (4) Removal of preliminary recycling studies to sharpen the manuscript's focus on the core contributions—namely, the densification process, electrode optimization, and comprehensive electrochemical performance.

1) The author has suggested that the decreased resistance observed in EIS data implies that the PILG effectively establishes ion and electron transfer pathways within the electrode. However, the data provided in Fig. 2 such as SEM images might not be a full representation to evaluate whether PILG has been uniformly deposited even at the pores located in depth. This raises the question of whether the observed improvement is due to uniform deposition or the sum of localized enhancements of properties. Could the author share any perspectives on this matter? Hence, the work could be reinforced if the author could provide any additional data to verify the uniformity of PILG distribution across the electrode pores?

Response: We sincerely appreciate the reviewer's insightful comment. We fully agree with the concern that the morphological characterization of NMC811 (73.9%)–PILG in the original manuscript. In the revised manuscript, we have significantly expanded our investigations by systematically varying the active material content from 73.9 wt% to 86.7 wt% and 92.7 wt%. This allowed us to thoroughly analyze the influence of a well-regulated PILG phase on both the morphological and electrochemical properties of the composite electrodes. Comprehensive morphological, compositional, and 3D tomographic characterizations have been added in the new Figures 3 and 6, as well as Supplementary Figures 13 and 24. Selective results are also presented in **Figure R7** below.

Specifically, we have added top-view and cross-sectional SEM and EDS analyses of electrodes with varying active material contents (**Figure R7a–c** or Fig. 3). The images clearly show that the NMC811 secondary particles and the synthetic PILG boundary phases are well integrated across all compositions. The cross-sectional SEM images and elemental overlay maps reveal a uniform elemental distribution throughout the composites (**Figure R7b, c**, or Fig. 3c, f, i), highlighting the microstructural evolution induced by densification as the active material content increases.

In addition, EDS line scan analysis confirms that as the NMC811 content increases from 73.9 wt% to 86.7 wt% and 92.7 wt%, the Ni signal becomes increasingly continuous, reflecting a more densely packed particle network (Fig. 3j). Meanwhile, the C signal, though still prominent, becomes more confined to narrower regions, indicating the formation of a thinner but well-integrated conductive boundary phase.

To further investigate the 3D microstructure, we employed FIB-SEM tomography to reconstruct the internal architecture of the composite electrodes (**Figure R7d** or Fig. 3k). This technique enabled us to clearly segment and distinguish among the conductive PILG phase, the pore network, and the active material phase. The segmented 3D images directly visualize the multi-phase microstructure and the interconnected conductive PILG network, emphasizing its crucial role in maintaining effective charge transport within densely packed electrode architectures.

Figure R7. **a**, Top view and **b**, **c**, cross-sectional view after vertical sectioning of the composite pellets with elemental mapping of Ni, Mn, Co, C, F, S and O elements using energy dispersive X-ray spectroscopy (EDS). **d**, 3D visualization image of NMC (86.7 %)-PILG electrodes. Scale bars, 250 μm (**a**), 25 μm (**b**), and 5 μm (**d**).

2) Throughout this work, NCM811 is consistently utilized as the primary active material, likely chosen for its ability to achieve high energy density. However, the author employs NCM111 as the active material to evaluate recyclability. The reviewer is curious about the reasoning behind selecting NCM111 for this aspect of the study. Could the author elaborate on why NCM111 was deemed more suitable for recyclability testing? Furthermore, if the recycling of NCM811 is particularly challenging due to its structural instability, what alternative approaches or strategies could the author consider addressing this issue? More explanation in depth would further clarify the author's claims.

Response: We sincerely appreciate the reviewer's insightful question regarding the selection of NCM111 for the recyclability evaluation. Initially, we included a direct recycling study using NCM111 to investigate the structural role and functional contribution of the engineered boundary phase during the relithiation of end-of-life composite electrodes. NCM111 was chosen at that stage due to its relatively higher structural stability under repeated chemical and thermal treatments, which allowed us to isolate and assess the effects of the boundary phase without the added complexity and instability associated with the more reactive and degradation-prone NCM811 chemistry.

However, in the revised manuscript, we have intentionally removed the recycling component in order to sharpen the focus on the core innovation of this work: the geology-inspired densification strategy and its role in optimizing the performance of thick, dense composite electrodes. This decision was made to ensure clarity and coherence in the manuscript's scientific narrative, as the recycling work—while promising—requires a separate, dedicated investigation.

Our future work will focus on developing an in-situ relithiation strategy without breaking the integrity of the composite. It involves the co-design of synthetic boundary phases that not only enhance mechanical integration during cycling, but also facilitate efficient mass transport during solution-based relithiation processes.

3) Continuing with the discussion on recycling experiment, the author provides the cell performance of recycled NCM111 in Fig. 5(j). The reviewer believes that the claims regarding the eco-friendliness of the designed electrodes could be reinforced by including a comparison between the cell performances of a newly fabricated electrode and a recycled electrode. The addition of this point could confirm the sustainable aspect of the proposed electrode.

Response: We thank the reviewer for the valuable suggestion. While we agree that such comparisons could enhance the discussion of recyclability, we have decided to remove the recycling-related studies from the revised manuscript in order to sharpen the focus on the core innovation of this work: the densification strategy and its impact on the electrochemical performance and mechanical properties of thick composite electrodes.

In the revised version, we have added expanded studies on the optimization of composite electrode formulation and processing conditions, which have led to a significant increase in active material content and a reduction in porosity. These enhancements allow for a substantial decrease in the amount of binder and conductive additives, while still achieving excellent electrochemical performance across gravimetric, areal, and volumetric metrics. Simultaneous enhancement of gravimetric, areal, and volumetric performance, which collectively improves cell-level energy performance for practical applications.

4) The author states that the thick electrode with PILG exhibits the least swelling compared to its counterparts, indicating robust structural integration. The reviewer would appreciate further elaboration on any potential side reactions, the formation of a cathode electrolyte interphase (CEI), or pulverization, if observed. It would be helpful to obtain a more comprehensive view of the compatibility of this electrode with carbonate-based electrolytes.

Response: We sincerely thank the reviewer for this important and insightful comment. As the reviewer rightly notes, understanding structural degradation mechanisms, such as pulverization

and interphase formation, is essential for evaluating the long-term stability and electrolyte compatibility of thick electrodes.

To address this point, we conducted additional post SEM-EDS analyses on cycled electrodes. After 300 cycles, despite some increase in surface porosity (**Figure R8a**), the electrodes maintained good structural integrity, suggesting that the PILG phase effectively mitigates mechanical degradation. Cross-sectional SEM images revealed that most NMC811 secondary particles retained their morphology, though some pulverization was observed near the separator-facing surface (**Figure R8a, b**). Additionally, EDS line scans showed elevated oxygen signals near the surface compared to pre-cycling conditions, while nickel signals remained largely unchanged (**Figure R8c** or Supplementary Fig. 24). This suggests the possible formation of oxygen-rich species such as LiOH, Li₂CO₃, NiO, and others, likely due to cathode–electrolyte interphase (CEI) formation under prolonged cycling

Importantly, despite the presence of these localized degradation features, the overall electrode structure remained mechanically and chemically stable. We attribute this stability to the 3D conductive PILG network, which plays a crucial role in mitigating electromechanical degradation—a common issue in conventional thick electrodes. This is further supported by cross-sectional thickness measurements, which showed minimal thickness variations (only 1.04× to 1.08×) even after 200–300 cycles.

In this revised manuscript, we also optimized the composition of electrically and ionically conductive components, as well as the processing conditions, to further increase the active material content from 73.9 wt% to 86.7 wt% and 92.7 wt% (Supplementary Table 1). Notably, reducing the PILG boundary phase not only improved the energy density per total electrode due to the increased active material content, but also reduced side reactions between NMC particles, additives, and the electrolyte, leading to extended cycling performance (Fig. 6c).

These new findings have been incorporated into the revised manuscript and are supported by the new Figure 6d and Supplementary Figure 24.

Figure R8. Morphology of post-cycled NMC811-PILG cathodes. **a**, Top view and **b**, cross-sectional view of the cycled NMC811 (86.7 %)-PILG electrode, along with corresponding elemental mapping of C, F, Ni, Co, and O obtained by EDS. **c**, EDS line scan profile of the cycled sample (> 200 cycles at 1 mA cm^{-2}) cross-sectional surface after vertical cutting, highlighting the Ni and O elements. Scale bars, $100\ \mu\text{m}$ (**a**) and $25\ \mu\text{m}$ (**b**).

Reviewer #4 (Remarks to the Author):

Manuscript No: NCOMMS-24-66206

Title: Unveiling multifunctional synthetic boundaries in densified thick composite electrodes to advance battery solutions

Decision: Reject

The manuscript reported liquid-polymer based densification method to fabricate thick NMC electrodes in the form of pellets. The impact of densification and its mechanical properties were studied followed by electrochemical performances. The objective of fabricating a robust thick battery electrode possessing a reasonable electrochemical performance is quite interesting.

However, the manuscript lacks evidences to support the densification in terms of microstructural properties, cross-sectional analysis, ion trade-off at the interface, and prototype-level proof. In addition, the migration of polymer at such a high temperature is inevitable which raises serious

concerns over the homogeneity of ion boundary or particle contact. Therefore, the manuscript is not suitable to be considered and I recommend to be rejected.

Response: We sincerely thank the reviewer for the constructive feedback and insightful suggestions, which have greatly motivated us to conduct additional experiments and improve the overall quality of our revised manuscript.

In response to the concerns, we have significantly expanded our experimental validation. Specifically, we have:

- Optimized the composition of the electrodes and densification process to enhance the active material content from 73.9 wt % to 92.7 wt %.
- Conducted comprehensive morphological analyses, including top-view and cross-sectional SEM imaging.
- Performed 3D microstructural reconstructions using FIB-SEM to elucidate the internal architecture.
- Evaluated electrochemical performance across gravimetric, areal, and volumetric metrics to provide a holistic view of electrode behaviors.

Inspired by the natural process of pressure-solution creep observed in geological densification, our approach leverages uniaxial pressure and transient liquid phases to create localized solvothermal microenvironments, enabling the fabrication of inorganic–organic composite electrodes. Originally developed for sintering ceramic powders at low temperatures (<250 °C), this method is still at an early stage of adaptation for complex electrochemical systems. However, it shows significant potential in achieving both mechanical robustness and enhanced electrochemical performance.

Unlike conventional slurry coating techniques that rely heavily on solvents, our method uses only a small amount (1 to 5 vol %) of a transient solvent system such as DMF or acetone. These triggers localized dissolution of additives (e.g., polymers and lithium salts) followed by their re-precipitation on pore surfaces during the thermomechanical process, promoting densification without excessive fluidity. Although adding higher-viscosity ionic liquids could improve mobility at elevated temperatures, our process remains fundamentally different from solution-based methods and is best described as thermomechanical powder processing.

The key innovation of this work lies in providing a novel route to fabricate thick and dense composite electrodes that simultaneously achieve high gravimetric, areal, and volumetric performance. Increasing electrode thickness is a promising strategy for boosting device-level energy density, but it often reduces gravimetric capacity due to sluggish charge transport. Porous thick electrodes may deliver good gravimetric performance yet suffer from low volumetric energy density when porosity exceeds 40 percent in previously reported thick electrodes. Our approach strikes a balance by reducing porosity while increasing thickness, preserving comprehensive electrochemical performance across all three key metrics (gravimetric, areal, and volumetric performance).

We fully agree with the reviewer’s concerns regarding the homogeneity of thick and dense electrodes, as well as the need for prototype-level validation to support practical application. Although it is not feasible to overcome every challenge of scaling this novel processing method from lab-scale (low technology readiness level) to a practical application at large-scale (high technology readiness level) in one study, we have made promising progress.

Specifically, we have demonstrated the feasibility of fabricating high-performance, dense, and thick electrodes using a small-sized die (12.7 mm in diameter) for coin cells, showing consistent homogeneity (see characterizations via SEM images, EDS mapping, and 3D reconstruction in **Figures R9, 10**). As part of our ongoing efforts, we are scaling up the process using a larger die for pouch-cell fabrication. The composite electrodes exhibit good integration processed in a rectangular die (57 mm × 46 mm) even as the applied pressure is reduced from 400 MPa to 50 MPa (**Figure R16a, b**). Battery performance using the pouch-cell configuration is currently under development. In particular, we are investigating the integration of high areal capacity anodes, such as graphite composites with areal capacities of $>15 \text{ mAh cm}^{-2}$ (see some preliminary results in **Figure R16c-e**), to complement the thick cathodes demonstrated in this work.

To address the challenge of maintaining homogeneity beyond batch-level processing, we have assembled an interdisciplinary team comprising materials scientists, chemists, manufacturing engineers, and data scientists. Together, we are developing a continuous pilot-scale manufacturing process using hot roller pressing—a well-established metal forming technique. This roll-to-roll system will feature multiple rollers with controlled pressure and temperature, along with integrated in-line, nondestructive monitoring techniques to ensure quality control and structural consistency throughout production. We believe these continued efforts demonstrate both the scientific merit and translational potential of our densification strategy to practical applications.

The comments are as follows,
1. The migration of polymer binder is inevitable under Increasing electrode thickness emerges as a viable strategy for boosting energy density at the device level -processed electrodes obtained at 120 degrees. In such case, the homogeneity of ion-conductive boundary is not sufficiently validated.

Response: We thank the reviewer for raising this important point. The homogeneity of the composite and the ion-conductive boundary phases is indeed critical for ensuring both mechanical integrity and electrochemical stability, particularly in minimizing mechanochemical degradation during battery cycling.

Unlike conventional slurry coating techniques that rely on large volumes of solvent, our method uses only a small amount (1–5 vol%) of a transient solvent system, such as DMF or acetone. This initiates localized dissolution of additives (e.g., polymer binders and lithium salts), followed by re-precipitation onto pore surfaces during the thermomechanical densification process. This approach enhances densification without inducing excessive fluids. While the use of high-viscosity ionic liquids could further improve ion mobility at elevated temperatures, our process remains fundamentally distinct from solution-based methods and is more accurately described as a transient liquid-assisted thermomechanical powder processing technique.

To achieve optimal uniformity, we investigated several mixing strategies, including manual grinding with a mortar and pestle, ball milling, and the use of a centrifugal mixer. We found that a combination of manual grinding followed by centrifugal mixing produced the most homogeneous mixture. This was validated through comprehensive post-densification characterizations. For example, EDS elemental mapping of cross-sectional samples showed uniform distributions of Ni, C, and F (**Figure R9a**), indicating effective dispersion of both active and conductive components. Moreover, strain mapping under tensile loading revealed a uniform distribution of strain concentration within the localized conductive boundary phases (PILG),

suggesting good multi-phase integration and mechanical consistency (**Figure R9b**). Operando strain mapping further confirmed the mechanical stability of the composite electrodes during battery cycling, attributed to a more uniform distribution of lithiation–delithiation-induced internal strain across the entire electrode along both lateral and thickness directions (**Figure R9c**). Additionally, the ability of these thick and dense electrodes to maintain stable cycling performance over 100 to 300 cycles (**Figure R13**) indirectly supports their structural and compositional uniformity. In contrast, early trial samples with unoptimized mixing and densification conditions typically failed within the first 20 cycles due to the electromechanical degradation and charge transport limitations in thick and dense electrodes. Despite its promise in lab-scale demonstration, widespread commercialization of this new electrode fabrication process requires addressing several scientific and engineering challenges for both batch (for pouch cell) and continuous processes (pilot manufacturing or R2R auto-manufacturing).

Figure R9. **a**, Top view and **b**, cross-sectional view after vertical sectioning of the composite pellets with elementary mapping of Ni, Mn, Co, C, F, S and O elements using energy dispersive X-ray spectroscopy (EDS) on the fractured surface after breaking the pellet. **b**, Real-time full-field Y-direction and X-direction strain mapping via DIC analysis at a global strain of 0.80 % under a uniaxial tensile test (stretching along the Y direction). Scale bars, 100 μm (**b**). **c**, Operando x- and

y-strain mappings (via DIC analysis) showing the cross-sectional view of the composite electrode (360 μm thick) in the same region at different SOC during delithiation at 2.0 mA cm^{-2} without applying the external force. 500 μm in the optical image, and 50 μm in the strain mapping (c).

2. Considering the mass loading of above 10 mg/cm^2 and content of NMC, the method would lead to severe migration/agglomeration of polymer.

Response: We fully agree with the reviewer that maintaining electrode homogeneity becomes increasingly critical as the areal mass loading increases. By carefully optimizing the composition of each component (ratios of active material, CNFs, graphene, PVDF, and IL) and refining the processing conditions, we have achieved good control over the uniformity of thick and dense composite electrodes in our batch-to-batch fabrication process using a small-sized die (diameter of 12.7 mm). This uniformity is supported by EDS elemental mapping, strain distribution analysis, and extended cycling performance (e.g., 300 cycles) of electrodes with mass loadings ranging from 40 to over 100 mg cm^{-2} . However, as we transition to fabricating larger samples using a bigger die (e.g., 57 mm \times 46 mm) or continuous hot roller pressing, ensuring uniformity across the electrode becomes more challenging. In our following work for large-scale production, we are addressing fundamental scale-up issues such as: even dispersion of the transient phase throughout the powder mixture; uniform particle rearrangement and compaction; minimization of pressure-induced stress gradients; homogeneous heat transfer; and consistent removal of transient liquids during processing. To tackle these challenges, we are implementing in-line sensors for real-time monitoring of multiphysical properties, enabling more effective process control. Furthermore, we are developing both physical and digital twins of this manufacturing system to support process optimization and establish a closed-loop manufacturing framework, which is more efficient than the current trial-and-error method.

3. The microstructural properties of densified cathodes, cycled and fresh ones via appropriate cross-section method is necessary. FIB-SEM could be such tool to reveal the microstructural properties.

Response: We appreciate the reviewer's insightful comment. We fully agree that the cross-sectional method employed in the previous manuscript did not sufficiently support the microstructural analysis of the densified cathodes, especially in distinguishing the functional phases and demonstrating structural stability before and after cycling.

In response to the reviewer's suggestion and other related comments, we have added more SEM and EDS results from the top-view and cross-sectional view of both fresh and cycled electrodes with various active contents of 73.9 wt%, 86.7 wt%, and 92.7 wt% (Fig. 3a-i, and Fig. 6d). The selective results are shown here in the **Figure R10a-e**. Specifically, the NMC811 secondary particles and synthetic PILG boundary phases are well integrated into composites with varying active material contents. The cross-sectional images and element overlay maps clearly revealed a uniform elemental distribution across the entire composites. This analysis highlights the densification-induced microstructural evolution with varying active material contents. Moreover, the line scan analysis further confirmed that as the NMC811 content increases from 73.9 wt% to 86.7 wt% and 92.7 wt%, the Ni signal becomes increasingly continuous, reflecting a more densely packed particle network (Fig. 3j). Concurrently, the C signal remains evident but appears more

confined to narrower regions, indicating the formation of a thinner, yet well-integrated conducting boundary phase.

In addition, after long-term cycling, SEM-EDS analyses revealed well-preserved structural integrity with limited pulverization localized near the separator-facing surface and minor increase in surface porosity (**Figure R10d, e**). Moreover, EDS line scans showed elevated oxygen signals near the surface compared to pre-cycling conditions, while nickel signals remained largely unchanged (Supplementary Fig. 24). This suggests the possible formation of oxygen-rich species such as LiOH, Li₂CO₃, NiO, and others, likely due to cathode–electrolyte interphase (CEI) formation under prolonged cycling.

Furthermore, we employed FIB-SEM tomography to reconstruct the 3D microstructure of the composite electrodes (Fig. 3k and **Figure R10f**). This analysis enabled us to clearly segment and distinguish among the conductive PILG phase, pore network, and active material phase. The segmented 3D images provide direct visualization of the multi-phase microstructures and interconnected conducting PILG phase within the composite, highlighting its role in enabling effective charge transport even in dense architectures.

Figure R10. Morphology of pre- and post-cycled NMC811-PILG cathodes. a-c, Top view and cross-sectional view of densified composites, along with corresponding elemental mapping

of C, F, Ni, Co, and O obtained by EDS for NMC811(86.7 %)-PILG, **d, e**, Top view and cross-sectional view of the cycled NMC811 (86.7 %)-PILG electrode, along with corresponding elemental mapping of C, F, Ni, Co, and O obtained by EDS. **f**, Demonstration of FIB-sliced SEM images and their corresponding 3D reconstructions, illustrating the integrated multi-phase composites and segmented conductive boundary phase. Scale bars, 50 μm (**a**), 10 μm (**b**), 25 μm (**c and e**), 100 μm (**d**), 5 μm (**f**).

4. The cross-section image depicted in Fig. 2b does not support the study.

Response: We thank the reviewer for this valuable comment. In the revised manuscript, we have replaced this image with more comprehensive studies of composite electrodes with three different active material contents, which have been added to the new Fig. 3 in the revised manuscript.

Furthermore, post-cycling SEM–EDS analysis revealed that the composite electrodes maintained good structural integrity after extended cycling, with only minor surface porosity and limited pulverization near the separator-facing side (as shown in the new Fig. 6). EDS line scans showed increased oxygen signals near the surface, suggesting the formation of oxygen-rich CEI species such as LiOH, Li₂CO₃, or NiO (new Supplementary Figs. 24).

We believe these improvements offer a more convincing validation of our composite design strategy and better support the conclusions of the study.

5. The Li enriched boundaries across the cathode particles offer additional supply of Li-ions. The relative estimation of Li-ions that were available from electrolyte and poly(ionicliquid)-electrode is missing.

Response: Thank you for the reviewer’s comments. We agree that Li-enriched conductive boundaries enhance charge-transport kinetics and improve battery performance. More importantly, our geology-inspired low temperature densification approach enables the seamless integration of diverse components into a highly compact form without thermal degradation. Therefore, it can incorporate various conducting components (electrically conductive carbon additives, ionically conductive PILG mixed phase) into synthetic conducting boundary phases that overcome transport limitations in thick, dense architectures. As a result, our composite electrodes—with areal mass loadings of 40 to 190 mg cm⁻², relative densities of 85 % to 87 %, and active-material contents of 73.9 % to 92.7 %—deliver simultaneously enhanced gravimetric and volumetric performance that have not been achieved in previously reported electrodes (**Figure R11a** or Fig. 6e). The estimation of Li sources incorporated in our composite electrodes with different active material contents of 73.9 wt%, 86.7 wt%, and 92.7 wt% are summarized in Supplementary Table 3 and 4).

As part of our future investigation, we have optimized the composite electrodes to further increase the relative density from 86% to above 92%, thus delivering a high volumetric capacity of 607 mAh cm⁻³ at 0.05 C (**Figure R11b**). To the best of our knowledge, this represents one of the highest volumetric performance reported for thick electrodes. Our results emphasize that electrode performance is governed by microstructures and the conducting phase in our dense and thick NMC811–PILG composite system. Simply adding more lithium salts into the electrode system cannot achieve such promising trade-off electrochemical performance across all three key metrics.

These investigations provide more insight into high-energy-density composite cathodes for the next-generation lithium batteries.

Figure R11. a, Comparison of volumetric and gravimetric capacities of our densified thick electrodes with various architected thick electrodes reported in the literature. **b**, Cycling performance of the NMC811(86.7%)-PILG electrode (Thickness: 249 μm ; Relative density > 92 %)

6. The authors mentioned a trade-off strategy, which must be accounted on the basis of ion-concentration in electrolyte and electrode.

Response: We appreciate the reviewer’s insightful comment. Indeed, the trade-off strategy is fundamentally governed by the distribution and concentration of ions within both the electrolyte and the electrode. In thick and dense electrodes, achieving high areal and volumetric capacity often comes at the cost of ion transport limitations. In our approach, the polymer-in-liquid-gel (PILG) phase forms Li-enriched boundaries at particle interfaces, serving as a charge “reservoir” like a solid-state electrolyte. These Li-rich interfacial pathways enhance local ionic conductivity and help mitigate concentration gradients that otherwise impair charge transport in high-mass-loading electrodes. Moving forward, we will leverage this PILG-mediated boundary design to explore true all-solid-state battery configurations. By tailoring the composition and processing of the boundary phase, we aim to develop electrodes that combine the mechanical robustness via our densification method with the safety and stability benefits of (quasi-) solid electrolytes (**Figure R12**, *Materials Today Energy* (2025), 49, 101829).

Fig. R12. Electrochemical performance of half-cell assembled with PILG phase-based oxide electrolytes (LATP-PILG). Rate performance and prolonged cycling at 0.1 C and 55 °C.

7. The content of NMC811 in densified electrodes is low about 81% considering commercial NMC811 electrodes (wet-processed) about 94%. There is a large gap in the active content among the above conditions.

Response: Thank you for pointing this out. By optimizing the composite electrode formulation and densification process, we successfully increased the active material content from 73.9 wt% in the original manuscript to 86.7 wt% and 92.7 wt% in the revised version. Achieving high active material loading in thick, dense electrodes while maintaining excellent electrochemical performance presents significant scientific and engineering challenges. Our transient liquid-assisted densification strategy enables the fabrication of high-density, thick composite electrodes, which are not attainable through conventional slurry-coating techniques.

As the active material content increases, ionic and electronic transport become more constrained due to limited percolation pathways within the composite matrix. Attempts to raise the active material fraction often result in poor electrochemical performance due to increased tortuosity and reduced effective transport channels. To overcome these limitations, we strategically employed transient liquid-assisted pressure-solution creep to form an interconnected, conducting polyionic liquid gel (PILG) phase. This PILG phase establishes a continuous 3D charge transport network. By carefully tuning the composition, morphology, and processing conditions of the PILG phase, we increased the active material content to 92.7 wt% through a compositionally adaptive densification process that regulates pore filling, particle connectivity, and interface contact under moderate uniaxial pressure.

This progressive enhancement in active material content directly translated into improved battery performance. As shown in **Figure R13** (new Figure 6a–c, and Supplementary Tables 1–4), increasing the active material fraction led to significantly higher volumetric capacity. Notably, the 92.7 wt% electrode delivered a volumetric capacity of 497 mAh cm⁻³ at 1 mA cm⁻², outperforming electrodes with lower loadings. To assess long-term performance, we selected a balanced electrode with 86.7 wt% active material content, which retained a capacity above 121.8 mAh g⁻¹ after 300 cycles (Figure R13c). While this retention is lower than that of slurry-coated thin-film electrodes, it is nevertheless promising given the significantly higher electrode density (porosity < 8–14%) and thickness (> 200 μm), enabled by improved charge transport through the synthetic conducting boundary phases.

Importantly, our thick, dense electrodes offer substantial advantages over conventional porous thin films in practical battery applications. As active material content increases, the required mass of passive components per ampere-hour (Ah) decreases, thereby enhancing cell-level energy density (**Figure R13d**, new Figure 6f, and Supplementary Tables 8–10). Simultaneously, the total volume required per Ah is reduced, demonstrating strong potential for space-constrained applications (**Figure R13e**, new Supplementary Figure 26, and new Supplementary Tables 8–10).

In summary, we have added new electrochemical results for composite electrodes with varying active material contents in the revised Figure 6a–c, f, as well as Supplementary Figures 23 and 26. Morphological characterizations of these electrodes are presented in the new Figure 3, Figure 6d, and Supplementary Figures 13 and 24 of the revised manuscript.

Figure R13. Battery performance of densified composite electrodes with various active material contents. **a**, Rate performance of NMC811-PILG electrodes with various active material contents of 73.9 %, 86.7 %, and 92.7 % (thickness: >150 μm , relative density: 86-88 %). **b**, Cycling performance of various NMC811-PILG electrodes at a current density of 1 mA cm^{-2} (thickness: >150 μm , relative density: 86-88 %). **c**, Long-term cycling of the NMC811 (86.7 %)-PILG electrode (thickness: 215 μm , relative density: 86%). Red triangles indicate the cycles in which a new Li foil was replaced. **d**, Comparison of the mass per ampere-hour of various active and passive components between slurry-coated thin-film electrodes and our NMC811-PILG electrodes with different active material loadings (excluding lead and packaging materials). **e**, Comparison of the volume per ampere-hour of various active and passive components between slurry-coated thin-film electrodes and our NMC811-PILG electrodes with different active material loadings.

8. Porosity estimation through calculation is theoretical. Since, the dense or thick electrodes properties are dependent on the porosity, the authors must provide evidence based on tomography.

Response: We sincerely appreciate the reviewer's insightful comment. As noted, theoretical porosity estimation is a commonly used approach; however, we agree that such estimations may not fully capture the complexity of multi-phase composite systems like ours. To address this limitation and provide direct experimental validation, we conducted FIB-SEM tomography.

Specifically, we acquired 400 serial cross-sectional images using a Ga⁺ ion beam with a slice thickness of 50 nm and reconstructed a 3D volume with dimensions of 28 μm × 18 μm × 20 μm (**Figure R14**). From this 3D dataset, we performed voxel-based segmentation to differentiate the active material, the conductive PILG phase, and pore regions. The reconstructed 3D morphology reveals a highly interconnected network of the conductive boundary phase percolating through the densely packed NMC811 particles. The volume fractions of each phase are summarized in Supplementary Table 5.

The experimentally determined porosity from the segmented 3D volume was 8.42%, which is notably lower than the theoretical estimate of 12.4% obtained via rough calculation (Supplementary Text). Additionally, we independently estimated the volume fraction of the conductive PILG phase using thermogravimetric analysis (TGA) deconvolution, which yielded a value of 30.22 vol%. This result closely aligns with the FIB-SEM-derived value of 34.72 vol% (including porosity), as shown in Supplementary Table 5. We also attempted micro-CT to reconstruct a larger 3D volume; however, the limited resolution (8 μm) was insufficient to capture detailed information about pores and boundary phases.

Together, these complementary methods reinforce the validity of our porosity evaluation and confirm the structural integrity of the dense composite electrodes.

Figure R14. Demonstration of FIB-sliced SEM images and their corresponding 3D reconstructions, illustrating the integrated multi-phase composites and segmented conductive boundary phase

9. The tortuosity is an essential parameter for thick electrodes, the calculation of tortuosity for the thick NMC electrodes and related to performance should be provided.

Response: We agree with the reviewer that tortuosity is a critical parameter in thick electrodes, as it significantly influences charge transport through the pore network. As noted in our previous response, the porosity of our densified electrodes is very low (<9%), as determined by 3D tomographic reconstruction. In this low-porosity structure, charge transport occurs through both the conductive boundary phase and the remaining pore network (**Figure R14** and Supplementary Table 5). To clarify these charge transport pathways, we analyzed the 3D spatial distribution of the conductive boundary phase and residual pores within the composite electrode. The reconstructed

3D network of conducting phase is highlighted in red (**Figure R14**), revealing its interconnected charge transport pathways throughout the densely packed structure.

10. Also, it is hard to compare the electrochemical data represented as mAh/cm³, mA/cm² with the previous reports. Suggest to report in a better format.

Response: Thank you for raising this important point. The central novelty of our work lies in the formation of a conductive secondary boundary phase that facilitates continuous charge transport within thick, dense electrodes, enabling the simultaneous optimization of gravimetric, areal, and volumetric performance, which is rarely achieved in prior studies.

To more clearly highlight these advantages, we have revised the manuscript to present electrochemical performance across all three metrics including gravimetric, areal, and volumetric capacities in a more structured and comparative format. Specifically, we provide a detailed analysis of the relationship between electrode thickness and electrochemical performance for conductive-phase-rich NMC811 (73.9%)–PILG electrodes (**Figure R15a** and revised Figure 5c–e).

Building on this, we further benchmarked the cell-level performance of our densified thick electrodes, which span a wide mass loading range of 44 to 187 mg cm⁻², against a conventional slurry-coated electrode with a practical loading of 14 mg cm⁻² (Figure 5e; see Supplementary Text for energy density calculations). Our densified electrodes exhibited superior performance across all key metrics: gravimetric energy density of 362–632 Wh kg⁻¹; volumetric energy density of 704–1338 Wh L⁻¹; areal energy density of 34–189 mWh cm⁻². In contrast, the slurry-coated electrode delivered only 207 Wh kg⁻¹, 256 Wh L⁻¹, and 10 mWh cm⁻². These improvements of our composite electrodes are attributed to the enhanced charge transport kinetics enabled by the conductive boundary phases, combined with increased electrode density and mass loading, which together deliver superior electrochemical performance across all relevant metrics.

Furthermore, we compared our densified electrodes to previously reported thick electrodes, as shown in **Figure R15b** (revised Figure 6e and Supplementary Table 7). Our optimized electrodes demonstrate a superior balance between gravimetric and volumetric capacity, consistently outperforming literature values. For example, at comparable or even higher gravimetric capacities, our electrodes achieved substantially higher volumetric capacities (430–500 mAh cm⁻³ vs. <100–350 mAh cm⁻³ in most prior reports).

Figure R15. a, Comparison of comprehensive cell-level performance (including areal, volumetric, and gravimetric energy densities) between our densified electrodes (mass loading: 44–187 mg cm⁻²

²) and a slurry-coated electrode with a practical mass loading of 14 mg cm^{-2} . **b**, Comparison of volumetric and gravimetric capacities of our densified thick electrodes with various architected thick electrodes reported in the literature.

11. The prototype application is missing like full cell or against Li (Li metal battery) in a large format like punch cell.

Response: We appreciate the reviewer's comment regarding prototype-scale validation. Translating a novel electrode fabrication process from the laboratory to large-format cells requires significant development and optimization. To date, we have demonstrated the feasibility of producing high-performance, dense, and thick electrodes using a small circular die (12.7 mm diameter) for coin cells, achieving consistent structural homogeneity across multiple batches. As part of our ongoing scale-up efforts, we are adapting the process to a larger die or continuous R2R process for pouch-cell prototypes. Early results indicate that our composite electrodes integrate well in large rectangular die ($57 \text{ mm} \times 46 \text{ mm}$), even when the applied pressure is reduced from 400 MPa to 50 MPa (**Figure R16a, b**), demonstrating the robustness of our densification approach.

Pouch-cell performance testing is currently underway, and further optimization is in progress. Specifically, we are pairing our thick cathodes with high areal capacity anodes, such as graphite with areal capacities up to 17 mAh cm^{-2} , to establish full-cell configurations that better reflect practical device operation (**Figure R16c–e**). However, fabricating large-sized electrodes with excellent electrochemical performance (e.g., pouch cell) will require overcoming several scientific and engineering challenges. These include ensuring even dispersion of the transient phase across particle surfaces, achieving uniform powder rearrangement and compaction, minimizing stress gradients during pressing, maintaining homogeneous heat transfer, and controlling the uniform evaporation of transient solvents. Addressing these challenges is part of our ongoing research and will be critical to making this low-temperature process both cost-effective and compatible with the current battery manufacturing infrastructure.

Figure R16. **a**, Batch scaling of densification by increasing the size of laboratory dies, from a circular die with a radius of 12.7 mm to one with a radius of 76.2 mm. **b**, A custom rectangular die

(57 mm × 46 mm) and the corresponding densified sample. **c**, Charge–discharge curves of graphite anodes with a thickness of 271 μm. **d**, Cycling performance of full cells comprising NMC811 (86.7%)-PILG electrodes (thickness: 346 μm; relative density >90%) and graphite anodes (thickness: 271 μm; relative density >72%) at a current density of 1 mA cm⁻². **e**, Comparison of charge–discharge curves of full cells.

12. There are some errors like LiNi_{0.80}Mn_{0.1}Co_{0.1}O₂ is NMC811 but not NCM811.

Response: We thank the reviewer for pointing out this oversight. We have thoroughly revised the manuscript to ensure consistent and accurate terminology throughout. All instances of “NCM811” have been corrected to “NMC811” in the revised version.

REVIEWER COMMENTS

Reviewer #1 (Remarks to the Author):

The authors have revised this work carefully. It can be published in Nature Commun now.

Response: We sincerely thank the reviewer for all the comments and suggestions. We greatly appreciate your recommendation for publication.

Reviewer #3 (Remarks to the Author):

The authors have thoroughly addressed all the concerns raised by the reviewer in the revised manuscript, and it is now recommended for publication.

Response: We sincerely appreciate reviewer's previous comments and recommendation for publication.

Reviewer #4 (Remarks to the Author):

Manuscript ID: NCOMMS-24-066206

Title: Unveiling multifunctional synthetic boundaries for enhanced mechanical and electrochemical performance in densified thick composite electrode

Decision: Revise

Authors undertook the revision based on the comments raised and were addressed in the revised manuscript. However, there results related to electrode cross-section need more clarity.

1. The cross-section image of densified NMC cathodes is still not satisfactory. Suggest the authors to provide cross-sectional image of densified thick electrodes of various mass loading 44 mg/cm² to 187 mg/cm², which they have claimed.

Response: We sincerely appreciate the reviewer's constructive feedback. We fully agree with the concern and acknowledge that additional cross-sectional imaging was necessary to better demonstrate thick electrode fabrication while maintaining high density. To address this, we conducted further experiments using the 86.7 wt% NMC811 composite, fabricating electrodes with areal loadings ranging from 65 mg/cm² to 162 mg/cm². The corresponding cross-sectional SEM images and EDS elemental maps (Fig. R1) have been compiled and incorporated into the Supplementary Information (Supplementary Fig. 14). We believe the updated figure provides a more comprehensive and representative depiction of the structure of our thick, dense composite electrodes.

Fig. R1. Cross-sectional SEM images and corresponding EDS elemental maps of NMC811 (86.7 wt%)-PILG composites with varying mass loadings (65 mg cm⁻² to 162 mg cm⁻²). Secondary electron (a) and backscattered electron (b) images of fractured cross-sections were acquired at the same magnification after pellet breakage, while EDS elemental maps for C, Ni, F, Mn, O, Co, and S (c) were obtained at optimized magnifications for each sample. Scale bars: 100 μm.